# Characterizing the scale of regional landslide triggering from storm hydrometeorology

Jonathan Perkins[1], Nina Oakley[2,3], Brian Collins[1], Skye Corbett[1], W. Paul Burgess[4]

[1]U.S. Geological Survey, Geology, Minerals, Energy, and Geophysics Science Center, Moffett Field, CA, 94035, USA
[2]Center for Western Weather and Water Extremes, Scripps Institute of Oceanography, San Diego, CA, 92037, USA
[3]California Geological Survey, Santa Rosa, CA, 95405, USA
[4]California Geological Survey, Los Angeles, CA, 90013, USA

*Correspondence to*: Jonathan Perkins (jperkins@usgs.gov)

**Abstract.** Rainfall strongly affects landslide triggering; however, understanding how storm characteristics relate to the severity of landslides at the regional scale has thus far remained unclear, despite the societal benefits that would result from defining this relationship. As mapped landslide inventories typically cover a small region relative to a storm system, here we develop a dimensionless index for landslide-inducing rainfall, $A^*$, based on extremes of modelled soil water relative to its local climatology. We calibrate $A^*$ using four landslide inventories, comprising over 11,000 individual landslides over four unique storm events, and find that a common threshold can be applied to estimate regional shallow landslide triggering potential across diverse climatic regimes in California (USA). We then use the spatial distribution of $A^*$, along with topography, to calculate the landslide potential area (*LPA*) for nine landslide-inducing storm events over the past twenty years, and test whether atmospheric metrics describing the strength of landfalling storms, such as integrated water vapor transport, correlate with the magnitude of hazardous landslide-inducing rainfall. We find that although the events with the largest *LPA* do occur during exceptional atmospheric river (AR) storms, the strength of landfalling atmospheric rivers does not scale neatly with landslide potential area, and even exceptionally strong ARs may yield minimal landslide impacts. Other factors, such as antecedent soil moisture driven by storm frequency, and mesoscale precipitation features within storms, are instead more likely to dictate the patterns of landslide-generating rainfall throughout the state.

## 1 Introduction

Rainfall-induced landslides are a global hazard that result in thousands of fatalities (Petley, 2012; Froude and Petley, 2018) and billions of dollars in economic losses annually (Schuster and Fleming, 1986; Kjekstad and Highland, 2009). During the progression of a hazardous storm, shallow landslides, those occurring primarily within a soil-mantled hillslope, are often triggered by infiltrating rainwater that interacts with the shallow (typically less than 3 m) groundwater system to produce destabilizing pore water pressures (Reid, 1994; Iverson, 2000; Collins and Znidarcic, 2004; Bogaard and Greco, 2016) (Fig.

1). Over the past five decades, a growing recognition of rainfall-induced landslide hazards has led to a range of efforts in developing landslide warning systems that assess when these rainfall thresholds for slope failure might be exceeded using a variety of criteria (Campbell, 1975; Keefer, 1987; Baum and Godt, 2010; Kirschbaum and Stanley, 2018; Guzzetti et al., 2020, and references therein).

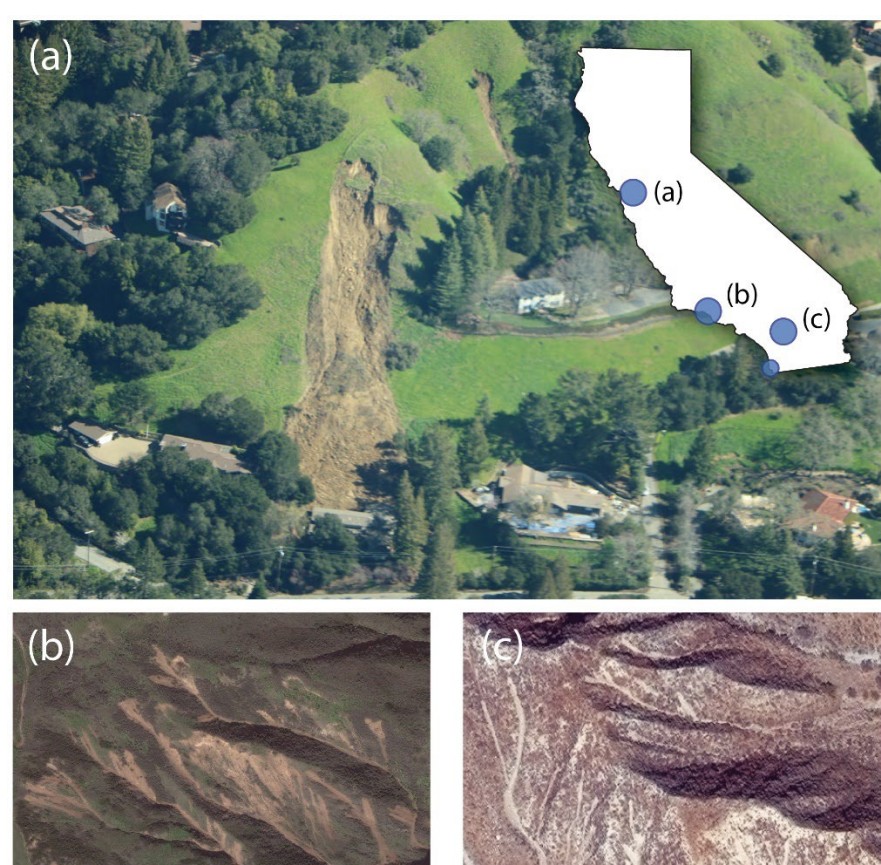

**Figure 1. Examples of landslides triggered by recent storms in California. (a) Aerial photograph of a home in California's East Bay region damaged by a landslide initiated during an atmospheric river storm on 06 February 2017. Photo taken by Brian Collins. (b) Worldview-2 imagery of landslides triggered by heavy rainfall on 10 January 2005, near the town of La Conchita. (c) Worldview-3 imagery of the southernmost San Bernardino Mountains north of Cabazon showing debris flows triggered by the 14 February 2019, atmospheric river storm that caused extensive damage across Riverside County. Inset shows a map of California with annotated circles corresponding to the respective panel. Unlabelled small blue circle corresponds to the location of the landslide inventory associated with a storm in April 2020 that triggered numerous landslides near the town of Encinitas (Fig. 3d).**


Whereas operational forecasting of landslides using numerical weather prediction remains rare (e.g., Guzzetti et al., 2020; Kong et al., 2020), a growing body of research suggests that distinct meteorological features at both the synoptic scale (~200 to 2000 km, multiple days) and the mesoscale (~2-200 km, minutes to hours) can exert a strong control on landslide occurrence and distribution and could potentially be used for landslide forecasting. For example, atmospheric rivers (ARs)

are synoptic features consisting of long filaments of enhanced water vapor in the lower atmosphere and are typically associated with mid-latitude cyclones that transport moisture poleward from the tropics to the mid-latitudes such as the west coast of the United States. They are a primary generator of precipitation in California (USA) and are typically measured and denoted by integrated water vapor transport (IVT) values exceeding 250 kg m$^{-1}$ s$^{-1}$ (Ralph et al., 2019). Combining news reports of landslide events going back nearly 150 years with an AR catalogue, Cordeira et al. (2019) showed that in

California's San Francisco Bay area 70-80% of reported landslide days occur in association with AR conditions. However, the authors also found that only 5-12% of ARs in their catalogue coincided with reported landslide days, leading them to suggest that other meteorological processes may have accompanied these storms to trigger the reported landslides. Similarly, Oakley et al. (2018) found that 60-90% of rainfall events exceeding published landslide-triggering thresholds in California over a 22-year period coincided with storms featuring ARs. At a smaller dynamic scale, mesoscale processes that operate

within synoptic storms and that are shorter-lived phenomena compared to ARs, can provide bursts of higher intensity rainfall that can also trigger abundant landsliding. Collins et al. (2020) found a tight spatial clustering between distributed shallow landslides from a 2018 storm in central California and the stalling of a narrow cold-frontal rainband (NCFR; a band of intense convective rainfall that can occur ahead of a cold front) that followed the passage of an atmospheric river over the region. Here, the timing of landslide triggering coincided with the NCFR rather than the AR, though rainfall associated with

the AR likely primed susceptible slopes for later triggering (Collins et al., 2020). Thus, storm characteristics at both the synoptic scale and mesoscale can play an important role in shallow landslide occurrence and distribution, and efforts to forecast landslide occurrence could benefit from assessing the likelihood of these meteorological processes occurring over particular landscapes.

Evaluating the magnitude of landslide hazard potential across the footprint of a given storm event requires some

way to estimate landslide triggering. Rainfall intensity-duration thresholds are a common empirical method used to assess the landslide potential for a given storm event (Cannon and Ellen, 1985; Keefer et al., 1987; Larsen and Simon, 1993; Guzetti et al., 2008). These relationships are typically calibrated regionally (or at a specific site near a rain gauge) and generally follow a power-law relationship where the triggering rainfall intensity declines exponentially with storm duration. This exponential relationship between rainfall intensity and duration for landslide-triggering implies that higher-intensity

storms require less rainfall depth to trigger landslides than lower-intensity storms, which is known to be related to the nonlinear soil moisture storage characteristics that dictate the transmission rate of infiltrating pore water (Green and Ampt, 1911; Richardson, 1922; Richards, 1931; Lu et al., 2011). In landscapes that do not rapidly drain between storm events,

antecedent rainfall may lower the amount of rainfall needed to reach critically unstable pore water pressure (Crozier and Eyles, 1980; Crozier, 1999; Glade et al., 2000; Bogaard and Greco, 2018).

80          Incorporating antecedent moisture into regional estimates of slope stability has taken several forms. Thomas et al. (2018) considered antecedent soil moisture and rainfall depth thresholds for driving positive pore water pressure in soil columns using physically based infiltration models (i.e., using the Richards equation; Richards, 1931). They found a nonlinear relationship between antecedent soil moisture and the necessary rainfall depth to generate pore water pressures that trigger shallow landslide initiation in California's San Francisco Bay region. The nonlinearity results from the shape of the

soil water characteristic curves: as soil saturates from dry to wet conditions, the soil hydraulic conductivity increases by several orders of magnitude (e.g., van Genuchten, 1980), resulting in increasingly fast transmission of pore water from the surface to the water table. The Antecedent Water Index ($AWI$) proposed by Godt et al. (2006) uses only rainfall data in a one-dimensional mass balance model initially derived by Wilson and Wieczorek (1995) that tracks theoretical predictions of soil water throughout a rainy season. This class of reduced complexity soil hydrologic models are commonly referred to as

"leaky barrel" or "tank" models, where rainfall immediately enters the model reservoir and drains at a rate proportional to the reservoir height. While $AWI$ does not directly incorporate the physical processes of rainfall infiltration into the soil surface (i.e., it does not use the nonlinear soil water characteristic curve relationships upon which the Richards equation is based), the model has nevertheless proven to capture the dynamics of a range of soil hydrologic processes. Where Wilson and Wieczorek (1995) calibrated their model to observed changes in pore water pressure for a landslide early warning

system in the San Francisco Bay area, Godt et al. (2006) calibrated $AWI$ to local measurements of soil water content and used an $AWI$ threshold as part of a decision tree to forecast landslide events in the Seattle, Washington (USA) region. Similarly, the Japanese Meteorological Agency used a three-tank model calibrated to a specific watershed to develop a Soil Water Index ($SWI$) that has been used to help establish rainfall-induced landslide thresholds across the country (Okada, 2001; Saito and Matsuyama, 2010). Additional examples of hydrometeorological thresholds used in various landslide

forecasting frameworks can be found in Mirus et al. (2024).

        Regional variability also plays a role in setting rainfall thresholds, and several studies have used various forms of normalization of rainfall and/or soil moisture variables to account for this variability (Cannon et al., 1985; Keefer, 1987; Wilson, 1997; Guzetti et al., 2008; Saito and Matsuyama, 2012; Peruccacci et al., 2017). Cannon (1988) normalized rainfall totals by the gauge-specific mean annual precipitation (MAP) to account for regional differences in triggering rainfall.

Wilson and Jayko (1997) later updated Cannon's maps using the "rainy day normal" ($RDN$ = MAP/number of rainy days) to further account for regional differences in triggering. They noted that that the recurrence interval of storm events is important in the equilibrium of landscapes. Marc et al. (2019) tested the efficacy of the 10-year recurrence, 48-hour rainfall anomaly ($R^*_{48}$) as a predictor of shallow landslide concentration in Japan and showed that a strong correlation exists between landslide concentration and the magnitude of the rainfall anomaly. For the same storm, Saito and Matsuyama (2012) showed

that normalizing the $SWI$ by its locally maximum value over the preceding decade also correlated with clustering of landslides.

Wilson and Jakyo (1988), Peruccacci et al. (2017) and Marc et al. (2019) all posited that landscapes may experience some degree of geomorphic tuning to extreme rainfall, and there are several potential reasons why either climatology or locally extreme rain might shape landscapes in ways that result in varying landslide triggering thresholds across climates.

For example, soil production and hence soil thickness can change with increasing mean annual precipitation (Richardson et al., 2019; Pelletier et al., 2015). This may have either a destabilizing effect through increasing soil thickness or a stabilizing effect by lowering hillslope gradients through enhanced diffusivity. Furthermore, root reinforcement of hillslopes is controlled by vegetation density (e.g., Schmidt et al., 2001), which also varies with precipitation (Nemani et al., 2002; Tao et al., 2016). Additionally, theoretical and numerical work shows that local rainfall intensity can alter long-term landscapes by

changing factors like drainage density and mean slope (Tucker and Slingerland, 1997), which in turn can lead to nonlinear increases in runoff (Carlston, 1963) that can subsequently drive shallow landslide and debris flow initiation. Thus, there is strong conceptual basis for the normalization of rainfall thresholds with respect to the regional climatology of their respective landscapes.

Quantifying the overall strength of storms that trigger shallow landslides also remains an ongoing challenge. One

way to characterize distributed storm strength is with the R-CAT scale (Ralph and Dettinger, 2012; Lamjiri et al., 2020), which uses three-day precipitation totals from distributed rain gauges to delineate broad categories of storm strength, from R-CAT 1 to R-CAT 4. This allows intercomparison of extreme rainfall events over the past century when sufficient gauge data exist. On a broader scale, the atmospheric river (AR) scale (Ralph et al., 2019) uses the magnitude and duration of the vertically integrated water vapor transport, IVT, to categorize the relative strength of atmospheric rivers on a scale of AR1 to

AR5 at a point. Although not a direct measurement of rainfall, this characterization avoids the dependence of storm impact prediction on site-specific rain gauge data. AR Scale values are suggested to correspond to a balance between beneficial and hazardous conditions, with an AR1 event representing primarily beneficial rainfall and an AR5 event representing primarily hazardous rainfall, yet the authors stress that these are only general guidelines and may often not be the case (Ralph et al., 2019). Although the R-CAT and AR scales allow for intercomparison of storm rainfall or IVT characteristics, they do not

specifically represent landslide hazard. For example, if an R-CAT 4 or an AR5 event occurs when soil conditions are dry, they might produce fewer (or no) landslides than if elevated soil moisture conditions were present preceding the storm event. These considerations warrant a more hazard-focused characterization of storms.

A primary aim of this study is to develop a simple hydrometeorological metric that can be used to delineate conditions consistent with regional shallow landslide occurrence and that can be mapped in space and time. Combining

aspects of both the rainfall anomaly approach of Marc et al. (2019) as well as Saito and Matsuyama (2012), here we develop a universal index for landslide triggering based on anomalous values of theoretical soil water derived from the hydrologic tank model of Godt et al. (2006) and Wilson and Wieczorek (1995), which we call $A^*$ (eqn. 1). To test the methodology, we use landslide inventories from four storms that span both arid and temperate regions of California, a vast and notably geomorphically and climatically diverse region. We show that in the case of our four study inventories, a threshold of $A^*$ can

be utilized to identify landslide events in both space and time, which bolsters the use of $A^*$ to broadly estimate regionally hazardous rainfall conditions outside the areas of mapped landslides.

To estimate the footprint of potentially hazardous (i.e., shallow landslide-inducing) rainfall across the state, we measure the distribution of hillslopes impacted by above-threshold $A^*$ for each storm to define a landslide potential area, *LPA*. A similar approach has been used effectively in studies quantifying the impacts of earthquake-induced landslides by considering both ground shaking and topographic metrics. For example, Marc et al. (2017) utilize seismic scaling relationships and topographic slope to delineate a cumulative landslide-affected area resulting from an earthquake, and Tanyas and Lombardo (2019) consider the role of both peak ground acceleration and topographic slope and relief to map coseismic landslide-affected areas.

We then investigate how the magnitude and spatial pattern of landslide-inducing conditions relate to meteorological process strength and spatial extent. We apply our methodology to a diverse set of nine impactful landslide-inducing storms across California from 2005-2021 (including the four calibration events). California's landscape encompasses 11 mapped geomorphic provinces distinct in their climatic and topographic characteristics (Jenkins, 1938), and therefore provides an ideal study area in which to evaluate the utility of our hazard index $A^*$ that represents theoretical estimates of anomalous soil water against highly variable climatological conditions. To examine how the strength of AR conditions relates to the severity of shallow landsliding, we compare the landslide potential area *(LPA)* with the AR Scale (Ralph et al., 2019), and show that while ARs are clearly important drivers of the events in our catalogue, antecedent conditions controlled by factors such as AR frequency rather than individual AR strength, as well as mesoscale features that often define the distribution of brief but intense periods of rainfall, appear to exert a dominant control on shallow landsliding and should therefore be assessed when examining patterns of landslide-inducing rainfall.

## 2 Methods

### 2.1 Development of dimensionless landslide index $A^*$

Here we develop a proxy for rainfall-induced shallow landslide potential by establishing a normalized index, $A^*$, that represents extreme values of the Antecedent Water Index (*AWI*) relative to its local climatology:

$$A^* = \frac{AWI}{AWI_{\text{RI}}} \tag{1}$$

where $AWI_{\text{RI}}$ is the value of *AWI* at a given recurrence interval (RI). This is similar to the normalized rainfall metric $R^*$ proposed by Marc et al. (2019) and also conceptually similar to the Normalized Soil Water Index pioneered by Okada (2001) and Saito and Matsuyama (2010); however, here we use the hydrologic tank framework of *AWI* since it does not rely on a specifically calibrated and more complex 3-tank model and has already been effectively utilized in applications of landslide forecasting along the U.S. west coast (Wilson and Wieczorek, 1995; Godt et al., 2006). Similar leaky bucket models have also been utilized to explore controls on monsoonal landslides in the Himalaya (Gabet et al., 2004; Burrows et al., 2023).

Importantly, this approach using $A^*$ to define a universal hydrometeorological landslide hazard index does not explicitly assess the susceptibility of individual slopes to rainfall-induced failure as is commonly done for physically based models of shallow slope stability (e.g., Montgomery and Dietrich, 1994; Baum et al., 2008). Rather, the normalization process is purposefully focused on a broader, regional scale. At this coarse spatial scale, we argue that distributions of $A^*$
illustrate overall patterns of hazardous rainfall and therefore help provide a framework for intercomparison of storms and the meteorological conditions associated with rainfall-induced landslides.

The *AWI* used in our study was formalized by Godt et al. (2006) to develop a landslide forecast system for Seattle, Washington (USA). The index provides a measure of theoretical soil water using a simple hydrologic tank model developed by Wilson and Wieczorek (1995). The tank model employs a mass balance where rainfall is immediately added to a reservoir
with a lower outlet that drains proportionally to the water level in the reservoir. In the model design, reservoir drainage does not occur until sufficient rain has fallen to completely fill soil pores bound by capillarity that restrict water flow. This filling parameter is termed $R_0$ and herein taken to be equal to 0.180 m (Godt et al., 2006) which is approximately the amount of water needed to bring a 1-m-thick loamy soil to field capacity. Once the seasonal rainfall depth exceeds $R_0$, the flux of additional soil water not bound by capillarity is modelled as follows:

$$AWI_t = AWI_{t-1}e^{-k_d\Delta t} + \frac{I_i}{k_d}(1 - e^{-k_d\Delta t}) \tag{2}$$

where $I_i$ is the rainfall rate added to the reservoir [m/hr], $k_d$ is a drainage constant that modulates the flux out of the system [1/hrs], $\Delta t$ is the time step [hrs], and the first term in the equation is the value of *AWI* [m] at the previous timestep (*t*-1) that has experienced drainage over $\Delta t$. Following rainfall, *AWI* decays back toward zero. The model assumes that $\Delta t$ exceeds the timescale required for infiltrating rainwater to integrate fully with the existing pore water in the soil. The model also resets at
the beginning of each water year to its initial condition (-$R_o$), which approximates the impact of processes like evapotranspiration that tend to dry soils from their field capacity back toward their residual moisture content. Application of this methodology to regions outside of Mediterranean climates like those within our study area may require further testing or a more direct incorporation of evapotranspiration processes. Whereas the constant $k_d$ in eqn. (2) may influence the local magnitude of *AWI*, for $A^*$ only the relative value is important for a given grid cell. For the case of the similar normalized soil
water index, the changing rate constant does not significantly impact the normalized index (Osanai et al., 2010), and we conduct a similar sensitivity analysis here. Lastly, although this model does not directly consider the physical processes of infiltration through the vadose zone into a shallow unconfined aquifer (e.g., Iverson, 2000; Collins and Znidarcic, 2004; Thomas et al., 2018), given that we are modelling changes in seasonal average moisture storage, the model simplifications are reasonable for a depth-averaged estimate of soil moisture, and Wilson and Wieczorek (1995) and Godt et al. (2006) both
show that the model can replicate changes in pore water pressure and soil moisture that have been used as part of landslide early warning systems in both northern California and Seattle, Washington, respectively. Here we consider *AWI* as reflecting a general mass balance of soil water and do not attempt to tie *AWI* to a specific measurable soil variable such as moisture content or pore pressure.

Estimating the spatial distribution of $AWI_{RI}$ (the recurrence value estimates of $AWI$) across our study area required
calculating an $AWI$ climatology using a gridded precipitation dataset from which we could estimate local $AWI$ values at
varying recurrence intervals. Here we used 4 km grids of six-hourly rainfall from the National Oceanic and Atmospheric
Administration (NOAA) California Nevada River Forecast Center (CNRFC) Stage IV Quantitative Precipitation Estimate
(QPE) (Seo and Breidenbach, 2002; Nelson et al., 2016; CNRFC, 2023a). These precipitation products are generated by
interpolating rain gauge data using the elevation-precipitation relationships established by the PRISM Climate Group
(PRISM, 2023). Unlike other NOAA River Forecast Centers, because of poor radar coverage in crucial mountainous areas in
California, the CNRFC does not incorporate radar data in their QPE (Nelson et al., 2016). From this archive of gridded
precipitation estimates we calculated six-hourly $AWI$ for Water Years 2004-2022 across the state of California. To then
calculate $AWI_{RI}$ values for each grid cell, we used a block maxima method (i.e., taking each annual maximum of $AWI$ for
each Water Year) to create generalized extreme value distributions from which we calculate local recurrence intervals along
each grid cell (e.g., Marc et al., 2019). Figure 2 illustrates the methodology for an example pixel in our study domain. Panel
(a) shows the 19-year time series of rainfall for a representative pixel in southern California, and panel (b) shows the $AWI$
model results calculated from the rainfall data, with annual maxima shown as open circles. These annual maxima are then
plotted as a histogram (panel (c)) from which an extreme value distribution can be fit (blue line). Recurrence intervals can
then be directly estimated from the best-fit distribution. To be discussed in Section 4.1, for this analysis we selected the 10-
year recurrence interval for $AWI_{RI}$, and the resulting grid was smoothed with a median filter to ensure continuity between
pixels.

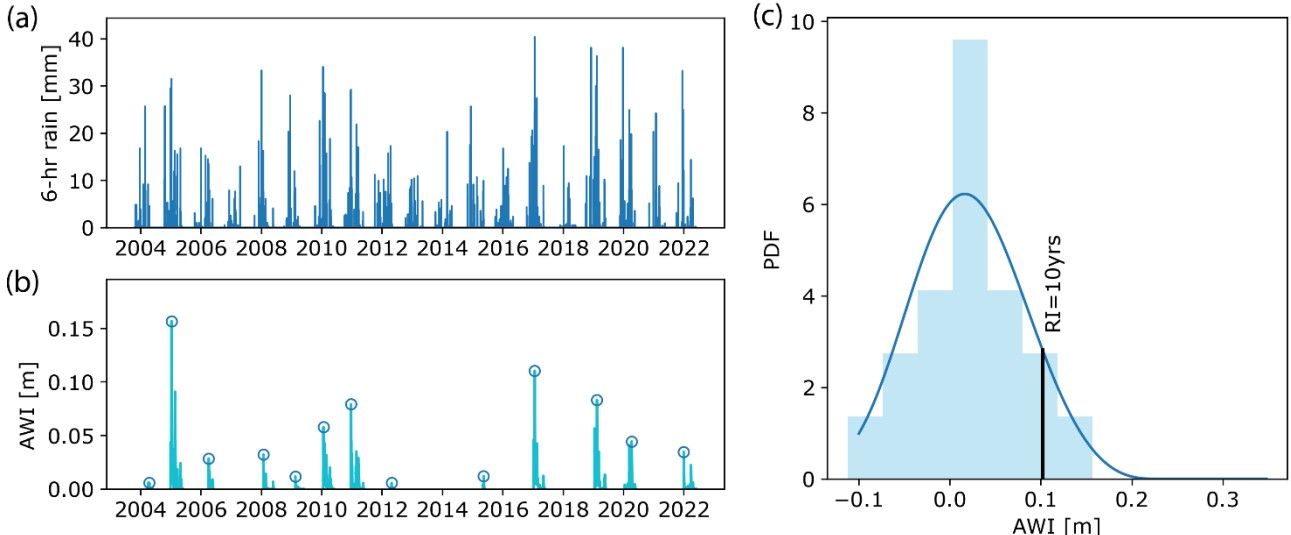

**Figure 2.** An example from southern California showing the methodology for calculating a climatology of *AWI* (b) using a 19-year record of six-hourly rainfall (a). *AWI* annual maxima are shown as open circles in (b), and years whose maxima are below zero (indicating that antecedent conditions were not met in this water year) are not shown. A generalized extreme value distribution is fit to a histogram of *AWI* annual maxima (c) from which any recurrence interval can be calculated. Here the 10-year recurrence value is shown as the bold black line in (c).

## 2.2 Determination of a common $A^*$ threshold using four rainfall-induced landslide inventories in California

We determine a common $A^*$ threshold using a series of four landslide-inducing storms that impacted different regions of California from 2005–2020 (Fig. 3). These events were chosen either because landslide inventories already existed or could be easily mapped from available satellite data. The four calibration events include a January 2005 storm that produced abundant landsliding throughout southern California (Corbett and Perkins, 2024a; Table 1), including the tragic La Conchita landslide that claimed 10 lives during the event (Jibson, 2006); a January 2017 storm that produced thousands of landslides in the East Bay hills of the San Francisco Bay area (Corbett et al., 2023; Thomas et al., 2017); a February 2019 storm that produced landslides both in the northern San Francisco Bay area as well as in southern California's Riverside County (Hatchett et al., 2020; Corbett and Perkins, 2024b); and an April 2020 storm that produced localized landslides and debris flows north of San Diego (CW3E, 2020; California Geological Survey, 2024)  (Table 1). These inventories together yield a total of 11,668 individual landslides.

To find a threshold that is consistent with all four storms, we first compare the maximum *AWI* for each landslide point (i.e., 11,688 points) during the passage of each storm against its background *AWI* value (as discussed below, we use the 10-year recurrence value). This serves as a simple test whether the triggering *AWI* is a constant threshold across different regions in California, which would plot as a horizontal line, or whether any threshold depends on the background *AWI* itself (sensu Cavagnaro et al., 2023). As the landslide spacing is small relative to the 4 km grid cells of the *AWI* dataset, we use the grdtrack function within the PyGMT software package (Uieda et al., 2021) that interpolates a precise value between neighboring grid cells. After identifying an acceptable common *AWI* recurrence interval for normalizing $A^*$ (see Results), we also examined the 19-year time series of $A^*$ across each inventory to illustrate the unique occurrence of these values throughout each of the four calibration storms.

## 2.3 Calculating the footprint of hazardous rainfall using landslide potential area (*LPA*)

One of the main goals of our study was to develop a methodology for both dynamically mapping conditions across the state consistent with distributed shallow landsliding, and for estimating the magnitude of potential landslide-affected area associated with a given storm. Our approach was to use $A^*$ as an index for distributed shallow landslide occurrence and then calculate the spatial distributions of maximum $A^*$ throughout the passage of a storm (Table 1). To do this, we identified the

time window bracketing the passage of each storm over land (typically on the order of 72 hours; Table 1) and then calculated the maximum of $A^*$ for each pixel in the domain. The landslide potential area (*LPA*) is then calculated simply as the area of hillslopes in our study area (units of km$^2$) with $A^* > 1$. To exclude flat terrain (i.e., not capable of shallow landsliding) and terrain covered in snow (where shallow landslides are unlikely), we created a mask of sloping terrain greater than 5° utilizing a 30 m SRTM-derived digital elevation model (DEM) (Farr et al., 2007), and also excluded grid cells with elevations greater than the typical winter snowline in the state (1024 m; see Hatchett et al., 2017). Here we also do not consider the bedrock-dominated deserts east of the Sierra Nevada where our soil moisture storage model for shallow landsliding is not strictly applicable. Although the 5° mask is a low threshold for shallow landslide producing hillslopes, we assume a conservative threshold given the relatively large DEM grid size compared to the size of typical shallow landslides. This yields a grid of shallow-landslide-prone terrain throughout the study area. To calculate *LPA* [km$^2$] we then interpolated the grid of $A^*$ maxima to the masked hillslope raster and calculated the area of hillslope cells with an $A^*$ maximum equal to or greater than our defined threshold. While here we do not propose that *LPA* specifically quantifies all areas impacted by landslides, instead we propose this approach offers a reasonable proxy of conditions consistent with observed shallow landsliding that can be used to coordinate potential landslide response.

## 2.4 Evaluating $A^*$ and *LPA* for a catalogue of recent landslide-triggering storms

We tested our analytical framework for regional shallow landslide triggering on a catalogue of nine landslide-inducing storm events in California since 2005, including the four calibration events described in Section 2.2 (Table 1). Whereas there are notable and well-documented landslide-inducing storms that occurred prior to 2005 in California, the gridded rainfall product we use in our analyses was not available prior to 2005 (Section 2.2.1). Thus, we were limited to evaluating only more recent storms. These storms were selected because they were either exceptionally large storms with a few well-documented landslide occurrences (essentially null events, e.g., October 2021), had mapped landslides with slightly less constrained timing (e.g., February 2005), or were storms with known reports of extensive landsliding but no available inventories (e.g., December 2005).

For each storm in our catalogue, we also examined the synoptic and mesoscale conditions using a variety of meteorological data. This included analysis of several meteorological variables such as geopotential height at various levels, integrated water vapor (IWV) and integrated water vapor transport (IVT), and upper-level winds from the ERA5 reanalysis dataset (Hersbach et al., 2020). We used NEXRAD weather radar data archived at the California-Nevada River Forecast Office (CNRFC, 2023b) and at the National Centers for Environmental Information (NCEI, 2023) to evaluate spatial patterns of rainfall in storms and to identify areas of short-duration, high-intensity rainfall associated with mesoscale features such as narrow cold frontal rainbands or thunderstorms, which are represented by high reflectivity values.

We calculated the AR scale value for each storm using the methodology of Ralph et al. (2019) at all ERA5 grid cells along the California coast for a time window spanning four days preceding the landslide event of interest; the AR scale

295      requires a minimum 72-hour window for calculation. We use the maximum AR scale at landfall in the State as representative of the AR scale of the event. This is common practice in reporting the magnitude of AR events affecting a broad region of interest (e.g., Center for Western Weather and Water Extremes, 2023a), but may differ from the AR scale value calculated at any individual landslide location. Most events affected multiple parts of the State, or the maximum AR scale at landfall corresponded to the location of one or more of the observed landslides. The exception is the April 2020 San Diego County

300      event. In this event, weak AR conditions were present in far northern California during the event window, but were irrelevant to the event itself, with no AR conditions present south of the San Francisco Bay. Thus, it was most appropriate to represent this event as 0, no AR. For the February 2005 Chino Hills event, AR1 conditions were present at a few grid points north of Point Conception, a far distance from the event but still in the broader Southern California region. While this event registered as having AR conditions on the AR scale, this event did not feature synoptic features consistent with an AR. Thus,

305      we rank it as AR1, but do not consider it as an AR in the synoptic features column.

| Event name and primary impacted regions in CA | Start Date (MM/DD/YYYY) | End Date (MM/DD/YYYY) | Synoptic Features | Mesoscale Features | AR Scale | *LPA* (km²) |
|---|---|---|---|---|---|---|
| January 2005. Transverse Ranges | 01/07/2005 | 01/11/2005 | AR, Closed low | Embedded convection | 3 | 11,950 |
| February 2005. Southern CA, Chino Hills | 02/18/2005 | 02/21/2005 | Cutoff low | Embedded convection | 1 | 6,160 |
| December 2005. Northern CA, Coast Ranges, Klamath, Sierra Nevada | 12/26/2005 | 01/03/2006 | AR | Ring-like band of moderate rainfall | 4 | 38,600 |
| January 2017. Northern CA | 01/08/2017 | 01/10/2017 | AR | Convective bands in Sierra Nevada, San Francisco Bay area | 5 | 25,750 |
| February 2017. Northern CA, San Francisco Bay area | 02/04/2017 | 02/08/2017 | AR | - | 5 | 5,920 |
| March 2018. Central CA coast; western Sierra Nevada foothills | 03/21/2018 | 03/23/2018 | AR | Narrow cold frontal rainband | 4 | 1,550 |
| February 2019. Statewide | 02/13/2019 | 02/16/2019 | AR | Convective bands in Sierra Nevada, embedded convection in Southern CA | 4 | 5,510 |
| April 2020. San Diego County | 04/07/2020 | 04/11/2020 | Cutoff low | Isolated thunderstorms | 0 | 1,620 |
| October 2021. Northern CA San Francisco Bay area, Sierra Nevada | 10/22/2021 | 10/25/2021 | AR | - | 5 | 60 |

**Table 1. Event catalogue of storms used in the analysis. The Synoptic Features column indicates whether the event featured a closed or cutoff low-pressure system, or an atmospheric river (AR), two synoptic-scale features commonly associated with impactful rainfall events in California (CA). The Mesoscale Features column indicates whether a mesoscale feature producing high-intensity rainfall (i.e., reflectivity >45 dBZ) was observed in radar imagery in the area where landslides were observed at the approximate time of landslide occurrence. "Embedded convection" refers to localized areas of high-intensity rainfall embedded within the broader storm system. Dashes indicate no observed features. Also indicated for each event are the measured Atmospheric River (AR) scale using the methodology of Ralph et al. (2019), and the calculated landslide potential area (*LPA*).**

## 3 Data: Meteorological characteristics of storms

The nine storms in our catalogue (Table 1) show a range of meteorological characteristics that caused rainfall-induced landslides. The January 2005 and February 2005 storms both impacted southern California; the January 2005 storm caused landslides along the coastal hillslopes and inland canyons of Ventura County (Jibson, 2006; Stock and Bellugi, 2011; Fig. 1) and the February 2005 storm produced hundreds of landslides in the Chino Hills region east of the city of Los Angeles (Prancevic et al., 2019). Both storms featured atmospheric rivers, with AR scale values of AR1 and AR3, respectively. They also exhibited embedded convection at the mesoscale, which can produce short bursts of high-intensity rainfall. Both events were also associated with cutoff- or closed- low pressure systems. Cutoff lows are mid-to-upper-level low pressure systems that are removed from the mean westerly flow and can result in persistent precipitation in a focused area (Oakley and Redmond, 2014; Barbero et al., 2019) thereby potentially affecting the resultant spatial distribution of landsliding. Localized zones of high-intensity precipitation during or in the vicinity of ARs figured prominently in several storms in our catalogue. For example, the December 2005 storm in northern California featured an extreme atmospheric river (AR4) and produced historic flooding and extensive landsliding across the region (Stock and Bellugi, 2011) including in the San Francisco Bay area, in the Klamath River region and in the Sierra Nevada.

The January 2017 and February 2017 events were part of a series of AR storms during the historically wet season of 2016-2017 in the San Francisco Bay area that produced over 9,000 landslides within the East Bay hills region alone (Corbett and Collins, 2023; Fig. 1). In the January 2017 storm in particular, convective bands of high-intensity precipitation were observed in both the Bay area and the Sierra Nevada foothills. In the March 2018 event, a stalling narrow cold-frontal rainband occurring immediately after the passage of AR conditions (AR4) produced abundant landslides over a section of the Tuolumne River canyon, west of Yosemite National Park (Collins et al., 2020).

The February 2019 AR storm showed evidence of convective bands in the Sierra Nevada (for reference, approximately 150 km east of the photo in Fig. 1a) and embedded convection in southern California, where historic flooding was observed in Riverside County (Hatchett et al., 2020) and hundreds of landslides occurred (Fig. 3c). The April 2020 storm was a cutoff-low pressure system. As the cutoff low passed over the San Diego region, isolated thunderstorms developed, producing high-intensity rainfall and triggering numerous landslides around the town of Encinitas (CGS, 2023; CW3E, 2020). This storm did not reach classification on the AR Scale.

Finally, the October 2021 storm consisted of an AR5 event on 24 October 2021 that that pummeled the U.S. West Coast and was the strongest AR to make landfall in northern California in the past 40 years during the month of October (CW3E, 2021). This storm led to flooding throughout northern California, in addition to isolated landslides in the northern California Coast Ranges and the northern Sierra Nevada.

# 4 Results

## 4.1 Determination of common $A^*$ threshold for shallow landslide-triggering conditions

The triggering *AWI* for landslides from each of the four calibration inventories in our catalogue (January 2005, January 2017, February 2019, and April 2020 storms; Table 1, Fig. 3a-d) varies with the background value of *AWI* for each location (Fig. 3e). Furthermore, the 10-year recurrence value of *AWI* ($AWI_{10}$) appears to serve as a common threshold (i.e., the 1:1 line) that over 97% of mapped landslides exceed across the four events. Whereas the January 2017, February 2019, and April 2020 landslide *AWI* points are closer to this threshold, the January 2005 event plots farther above the 1:1 line. While this appears to suggest that hillslopes with a higher $AWI_{10}$ may have a comparatively higher triggering threshold, evaluation of more landslide events across a broader climatic gradient is required to test this idea sufficiently. We thus take $AWI_{10}$ as the universal normalization parameter in the calculation of $A^*$ for this analysis (eqn. 1).

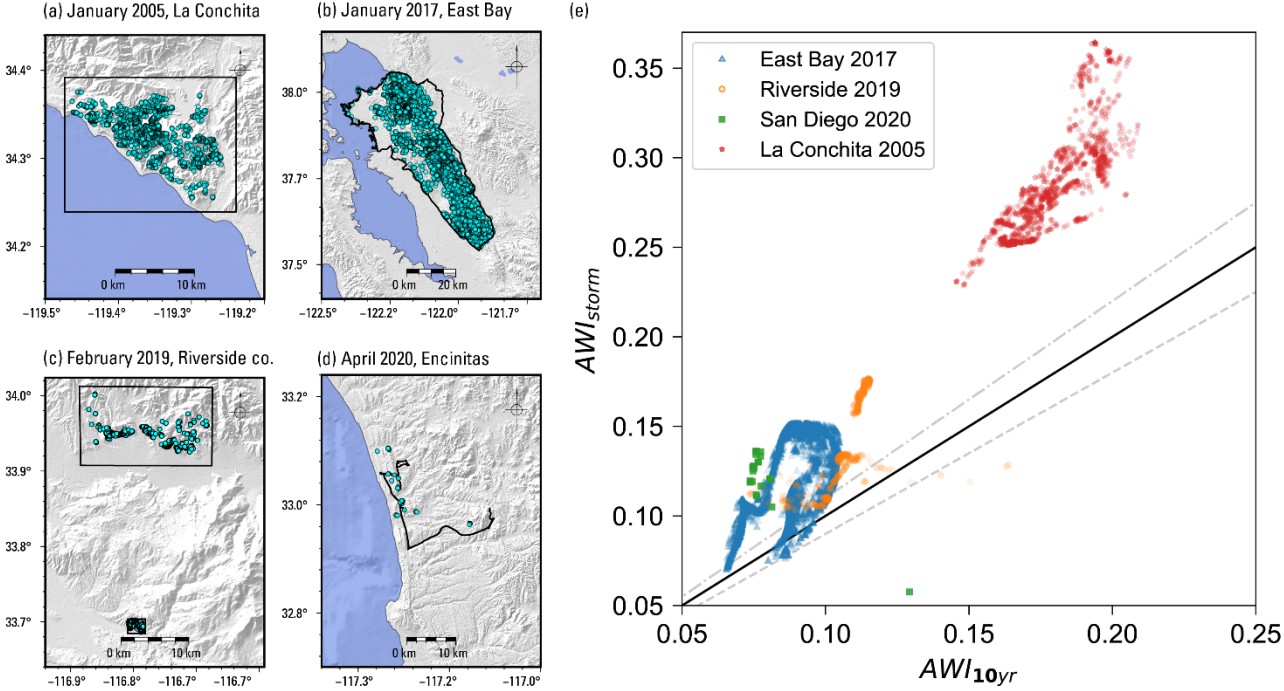

**Figure 3. Landslide inventories (a-d) used to estimate a reasonable antecedent water index (*AWI*) recurrence threshold above which most landslides occurred. Black boxes and polygon in panels (a)-(c) represent the landslide mapping bounds for each event, and black line in panel (d) represents the GPS tracks from CGS field mapping campaign. Panel (e) shows a plot of peak *AWI* modelled during the storm windows interpolated to each landslide point (y-axis) against the 10-year recurrence *AWI* at each point (x-axis). A constant threshold across regions would plot as a horizontal line. Over 97% of the mapped landslides plot above their 10-year recurrence value (the 1:1 line). Dashed lines are the 0.9:1 and 1.1:1 lines. Shaded relief for (a)-(d) derived from NASA SRTM 30 m DEM (NASA, 2013).**

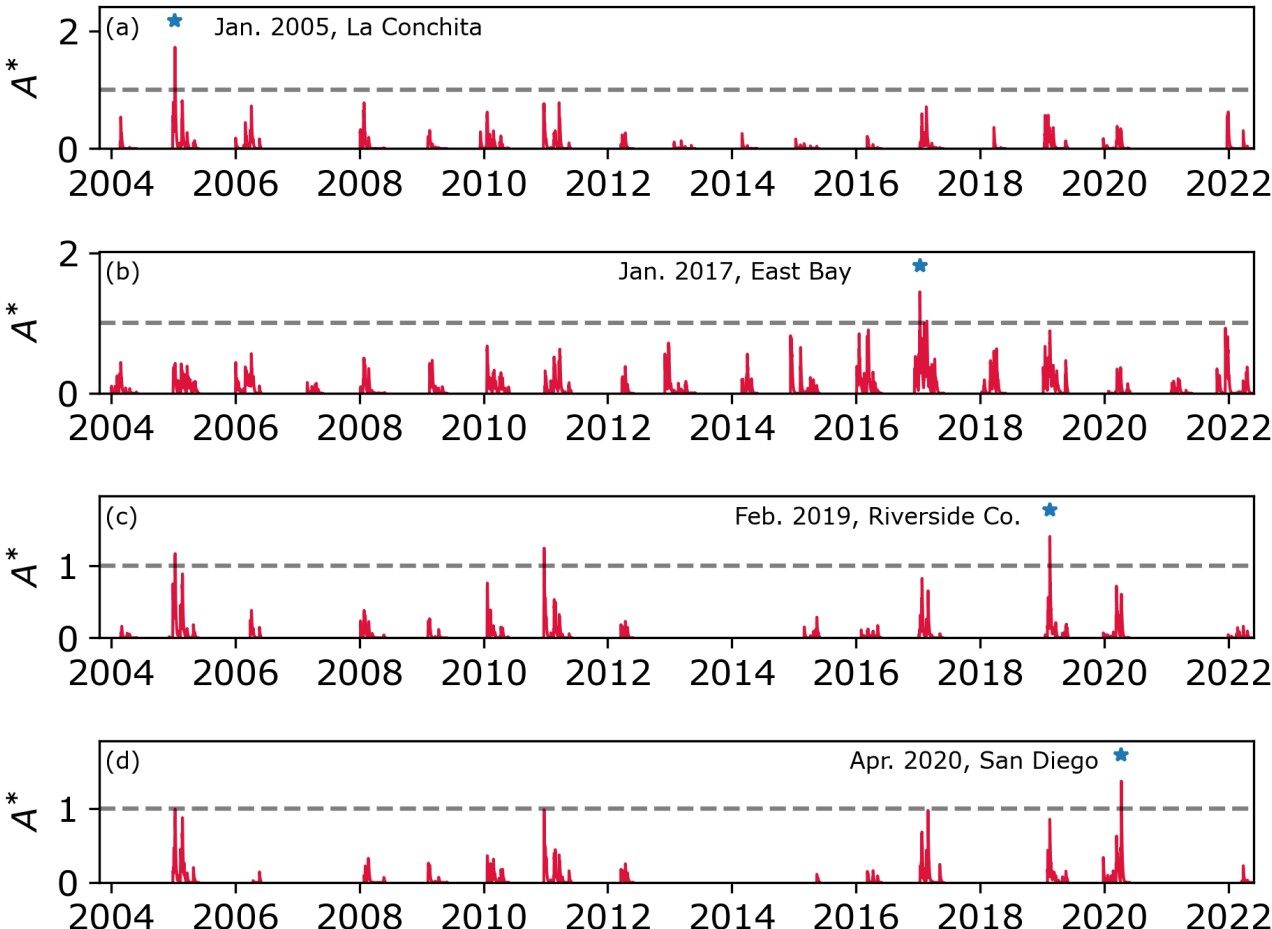


**Figure 4. Time series of median $A^*$ within a box surrounding each landslide inventory shown in Fig. 3. Dashed black line corresponds to a threshold value of 1, equivalent to the 10-year recurrence value of modelled antecedent water index (*AWI*) (eqn. 2) at each site. Approximate landslide timing (black star) corresponds to the maximum value of $A^*$ across each respective time series. For the case of (b), where landslide observations have been commonplace, no similar instance of extensive landsliding (e.g.,**

**Coe and Godt, 2001) has occurred during the modelled interval indicating no false positives. For the case of (c), the above-threshold peak from December 2010 corresponds to a massive regional storm event that produced numerous landslides and flooding in the region, and the 2005 peak corresponds to the same landslide-inducing storm described in (a), which also produced landslides in Riverside County (see discussion in section 4.1).**

Because our definition of $A^*$ utilizes a high storm recurrence interval (i.e., 10 years) $A^*$ values above 1 are, by definition, rare. Yet we nevertheless find value in looking at the 20-year time series of $A^*$ across each of our landslide

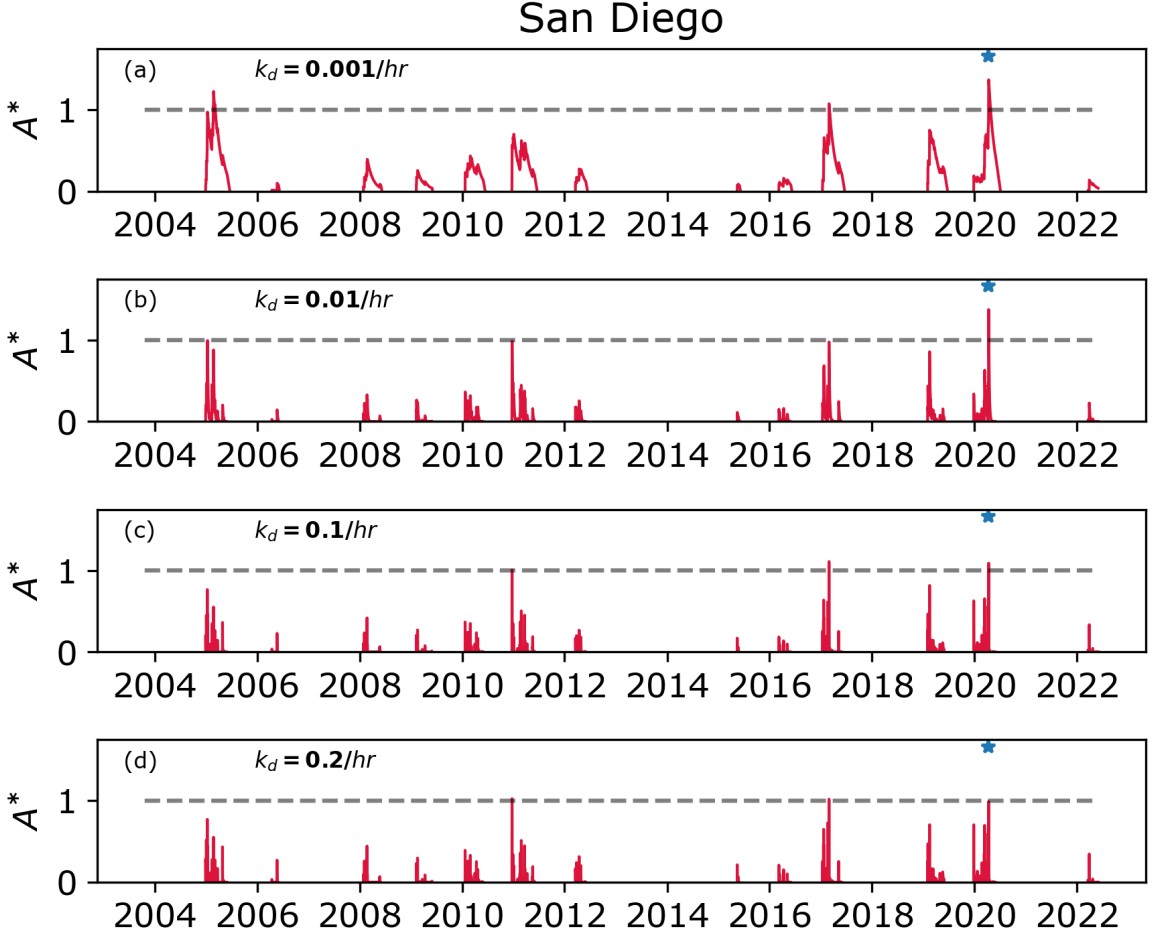

**Figure 5. Plots showing the sensitivity of $A^*$ to the drainage coefficient $k_d$ used in the *AWI* model (Eqn. 2) for the San Diego example shown in Fig. 4d. Red lines show the $A^*$ time series for each $k_d$ value ranging from 0.001 hr$^{-1}$ (a) to 0.2 hr$^{-1}$ (d). Panel (b) is**
**the same data as shown in Fig. 4d. Blue star denotes landslide timing in the April 2020 event, and grey dashed line represents the threshold $A^*$ value of 1 (e.g., Fig. 3). In this example, landslide timing is at or near the threshold value across the range of $k_d$; however, overall peaks in $A^*$ are broader for slower-draining soils (e.g., (a)) and narrow considerably with increasing drainage rate as the effect of soil storage declines. Thus, peaks in $A^*$ at these drainage rates depend more on instantaneous rainfall intensity and less on multi-day accumulation which featured prominently in the April 2020 storm (Table 1).**


calibration sites for which we have consistent rainfall data. For each of the four calibration events, we find that the landslide-inducing storm exhibited the largest peak in $A^*$ across their respective 20-year histories (Fig. 4). At a minimum, this implies

setting an $A^*$ threshold of 1 produces no false positives for each site, with the possible exception of the February 2019 Riverside area (Fig. 4c). Here there are two additional above-threshold peaks in the ~19-year climatology. The early peak

coincides with the January 2005 event, and while we do not have landslide mapping from this event in this region, landslides were indeed reported in surrounding Riverside County from this event (e.g., Los Angeles Times, 2005). Similarly, the second peak in $A^*$ occurred in December 2010 and triggered landslides and debris flows across southern California, including in Riverside County, leading to a request for \$110 million in federal disaster relief for storm damage (FEMA, 2011). Thus, while we cannot corroborate these two events as producing landslides within the specific boxes due to a lack of

detailed landslide inventory information, at the local scale they may be considered true positives. In the case of the April 2020 storm in the San Diego region, we also show that while the pattern of $A^*$ is somewhat sensitive to the choice of drainage parameter $k_d$, the timing of the landslide event is captured over several orders of magnitude in $k_d$ (Fig. 5). This suggests that $A^*$ is relatively insensitive to $k_d$ within this region, however more work is required to fully assess how much $k_d$ can influence $A^*$ across landscapes since soil drainage rates are often highly spatially heterogeneous.

When considering false negatives (i.e., distributed shallow landslides associated with $A^*$ values less than 1), assessing their outcome becomes more difficult because we do not have detailed histories of landsliding (or absence thereof) at all four sites. However, for the East Bay hills in the San Francisco Bay area, (Fig. 3d, 4d), we do know that the regional distributed landsliding produced by the January 2017 and February 2017 storms (combined number of landslides > 9000) had not been previously observed since the winter of 1997-1998 (Coe and Godt, 2001; Corbett and Collins, 2023). Because

these storms occurred so closely in time, it is not possible to determine which of the January or February 2017 storms produced the majority of landslides (Fig. 4b), although both are known to have caused landslides. Notably, both events produced $A^*$ values exceeding 1 within the map area (Fig. 6e,h). Overall, we see that mapped landslides from each of these four calibration storms coincide with peaks in $A^*$ in both space and time, and that a common threshold value of $A^*$=1 based on a comparison to the 10-year climatology can be applied to discriminate the events from storms that occurred in these

locations and that did not produce widespread landsliding.

### 4.2 $A^*$, *LPA*, and the impact of atmospheric river strength

All nine storms in our catalogue show at least some patches of above-threshold $A^*$; however, the magnitude and spatial distribution of $A^*$ is highly variable (Fig. 6). The inter-quartile ranges of $A^*$ for above-threshold hillslopes mostly occur between 1.0 and 1.1, and do not markedly change with the area of impacted hillslopes (Fig. 7a). Both the January 2005 and

February 2005 storms show larger inter-quartile ranges of $A^*$ with higher absolute values, and interestingly, both storms occurred within two months of each other in the winter of 2005 and impacted the same regions within southern California (Fig. 6d, g). Both storms had embedded convection and favorable orographic conditions (Table 1), which can lead to locally high rainfall totals (Section 4.1). *LPA* values, which represent the total area of hillslopes experiencing above-threshold $A^*$ for each storm, vary by nearly an order of magnitude and range from approximately 60 km$^2$ in the case of the October 2021

event to just over ~38,000 km² in the case of the December 2005 storm that led to severe flooding and landslides across northern California (Figs. 6-7).

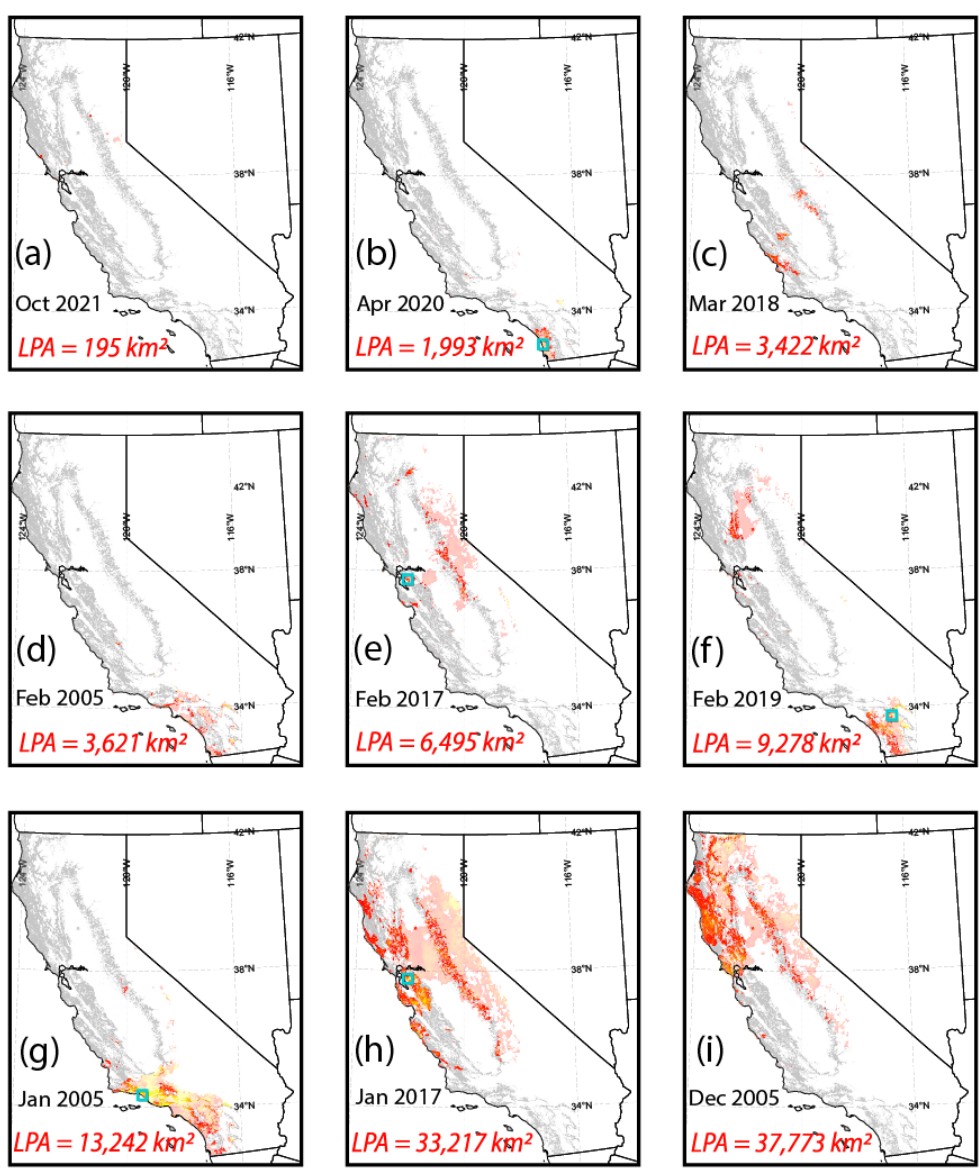

**Figure 6. Distributions of $A^*$ and resulting landslide potential area (_LPA_) for the nine landslide-inducing storms in our catalogue (Table 1). Panel numbers (a) – (i) are ranked in order of increasing _LPA_: (a) October 2021, northern California (CA); (b) March 2018, Central Coast and Sierra Nevada; (c) April 2020, Encinitas, (d) February 2019, Riverside County; (e) February 2017, northern CA; (f) February 2005, Chino Hills and eastern CA; (g) January 2005, La Conchita and southern CA; (h) Jan 2017,**

northern CA; (i) December 2005, northern CA. Hillslopes in our study region are shown in grey, and distributions of $A^*$ are shown as warm colors from $A^*$=1 (orange) to $A^*$=>1.75 (yellow). $A^*$ values outside of hillslopes are shown as semi-transparent, and approximate landslide inventory bounds are shown as teal squares. Topographic data derived from NASA SRTM 30 m DEM (NASA, 2013).

Given that most storms in our catalogue feature ARs, it is logical to investigate how the magnitude of shallow landsliding, as represented by *LPA*, compares to the magnitude of the associated AR conditions via the Ralph et al. (2019) AR Scale. Our results show that whereas the largest *LPA* events do generally correspond to large AR events, there is considerable variability between AR magnitude and landslide magnitude (Fig. 7a). For example, the three storms reaching AR5 (January 2017, February 2017, and October 2021) span the smallest *LPA* to the second largest (Table 1), indicating limited predictability of landslide hazard from measures of IVT alone. This comports with the findings of Cordeira et al. (2019) who find only a small percentage of reported ARs are associated with reported landslides in the San Francisco Bay area. One reason for this variability is precisely related to our basing $A^*$ on a model that accounts for antecedent soil moisture conditions. The AR scale does not incorporate any information on antecedent precipitation or soil moisture conditions that may precondition hillslopes and potentially affect subsequent landslide triggering.

Notably, when event *LPA* is plotted against the month in which the storm occurred, a more systematic relationship becomes apparent (Fig. 7b). Within our event catalogue, the largest landslide responses occur in late December and January,

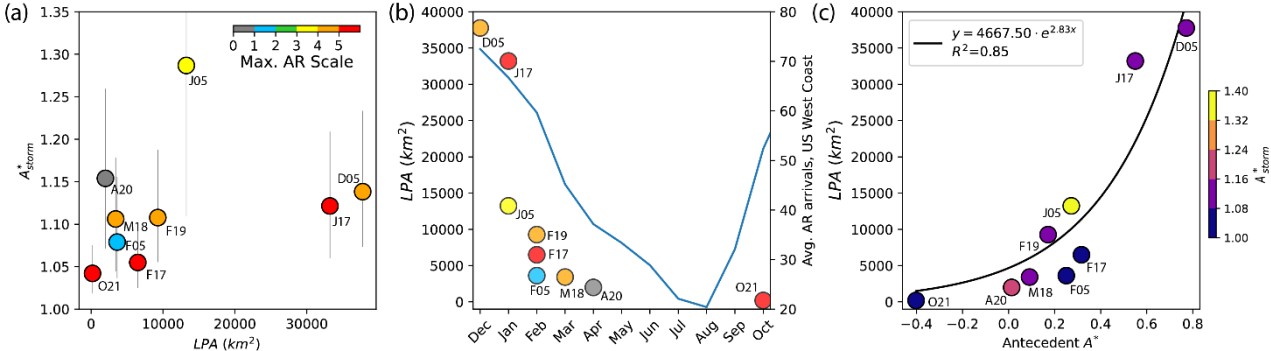

**Figure 7. Plots showing the relationship between $A^*$, landslide potential area (*LPA*), and the Ralph et al. (2019) atmospheric river (AR) scale. Panel (a) shows how the population of above-threshold $A^*$ varies with *LPA* and the AR scale (color). Dots show the median value of above-threshold $A^*$, and vertical lines show the interquartile range. While most events have median values somewhere close to 1, both the January 2005 and April 2020 events have higher median values above 1.15 and much larger interquartile ranges. This likely reflects either the strong orographic and/or convective nature of these two storms in southern California (see discussion). High AR-scale events exhibit both the highest and lowest values of *LPA* in our catalogue. Panel (b) shows *LPA* variation (left axis) with the time of year. Events in December and January have the highest *LPA*, with decreasing**

**impacted area (i.e., smaller *LPA*) later in the rainy season. Right axis shows the average annual AR arrivals along the U.S. west coast from reanalysis data (Mundhenk et al., 2016). Panel (c) shows the relationship between antecedent $A^*$ (the value of $A^*$ preceding a given storm window) for pixels that ultimately exceeded the $A^*=1$ threshold, and resultant *LPA*. The relationship is well fit by an exponential relationship (black line).**


and progressively decline throughout the year in an almost exponential fashion. Although the event catalogue is lacking in spring events relative to winter events, the overall apparent trend indicates that seasonal processes are at play that likely modulate the antecedent hydrologic conditions in landslide-prone hillslopes. This supports our use of a soil water balance

(i.e., *AWI*) anomaly-based metric for identifying landslide-inducing storms; soil moisture generally decreases in the spring months (March-April-May) as storms become less frequent (Figure 7b) and evapotranspiration increases with longer days and temperatures. Thus, *LPA* tracks well with storm frequency metrics such as the frequency of AR arrivals along the west coast of the United States (Mundhenk et al., 2016) which peaks in December and January and declines similarly to the monthly decline in *LPA* (Fig. 7b). This is similar to the January peak observed in the monthly frequency of historic landslide

days in the San Francisco Bay area region (Cordiera et al., 2019) as well as the peak in observed seasonal shallow landslide activity in the Pacific Northwest (Luna and Korup, 2022), indicating the role of soil moisture storage and groundwater conditions in driving the seasonality of regional shallow landslide activity (Luna and Korup, 2022). Because $A^*$ and *LPA* represent local extremes of soil water, this consistent trend across all events suggests that the observed seasonality in *LPA* persists across the state despite large differences in regional climate.

To test this relationship more explicitly, we examined whether storm *LPA* correlates with the degree of antecedent $A^*$ values for pixels that ultimately exceeded the landslide threshold during a storm event. Fig. 7c reveals a nonlinear relationship where the largest-*LPA* events in the catalogue tend to have higher antecedent $A^*$ values, and both early-season (e.g., October 2021) and late-season (e.g., April 2020) storms with low antecedent $A^*$ conditions exhibit comparatively low *LPA* values. This relationship is well fit by an exponential function ($R^2 = 0.85$), indicating that, perhaps unsurprisingly, the

degree to which the landscape is pre-conditioned by prior rainfall exerts a strong control on the area impacted by landslide-inducing rainfall. This result may therefore provide a causative link between the apparent relationship between monthly AR arrival frequency and the magnitude of landslide potential area (Fig. 7b), as frequent rainfall events across a region may keep the hydrologic mass balance in an elevated state more prone to exceed its local threshold should a comparatively strong storm system arrive.

## 5 Discussion and Conclusions

### 5.1 Effect of low antecedent moisture on large, early-season storm impacts

The October 2021 AR5 storm offers an important example of how low antecedent soil moisture can blunt the impacts of an exceptional storm producing record precipitation in California's highly seasonal climate (Fig. 8). This storm followed a year of drought and came uncharacteristically early for an AR of its magnitude (e.g., Ralph et al., 2019; CW3E, 2021). Because of this, soils were close to their residual moisture content. Despite generating a wide swath of highly anomalous 48-hour rainfall with $R^*_{48}$ values locally exceeding values of 2 from the San Francisco Bay area to the Sierra Nevada (Fig. 8a,b), no reports of major landsliding occurred outside of a few isolated events. Notably, Marc et al. (2019) report $R^*_{48}$ values exceeding 2 as an approximate threshold for what should lead to high-density distributed landsliding. In our calculation of $A^*$, the initially dry soil conditions at the storm onset that occurred only a few weeks into the rainfall season (beginning October 1 in California), contributed to a diminished distribution of $A^*$ and therefore little predicted landsliding (Fig. 8c). Thus, in Mediterranean climates where dry soils can mitigate the hazardous effects of anomalously high rainfall, consideration of soil storage is an important factor when using normalized thresholds for regional prediction of shallow landslides in soil-mantled hillslopes.

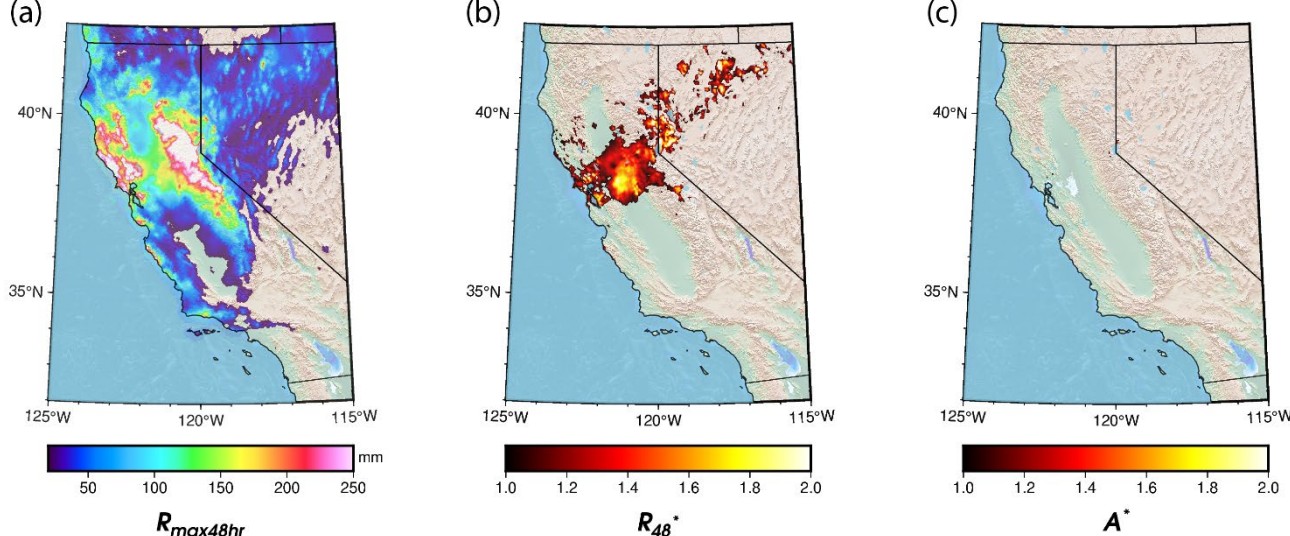

**Figure 8. (a) Map showing 48-hour maximum precipitation from the October 25th, 2021 Atmospheric River scale 5 (AR5) that struck northern and central California and produced widespread flooding but few landslides (Table 1). (b) $R^*_{48}$ metric of Marc et**

al. (2019) showing highly anomalous two-day rainfall totals for the region, calculated by taking the results of panel (a) and dividing by the 10-year recurrence 48-hour rainfall estimates from the NOAA Atlas-14 dataset (Perica et al., 2014). (c) Map of $A^*$ showing that despite anomalously high rainfall, few impacts from distributed shallow landslide occurrence may be expected. Shaded relief for all plots from NASA SRTM 30 m DEM (NASA, 2013).

**5.2 Dissecting the role of synoptic and mesoscale meteorological processes on landslide hazard**

Our study of a wide range of landslide-inducing storms allows evaluation of the role that storm characteristics might have on the distribution of landslides. We found that whereas AR presence is often associated with landslide events (e.g., Cordeira et al., 2019), the strength of ARs as measured by the AR Scale did not exert a significant control on the magnitude of landslide-triggering rainfall investigated here (Fig. 7a).

We also find that mesoscale features producing short-duration, high-intensity rainfall may play a more important role in dictating where shallow landslides and associated debris flows occur (Wooten et al., 2008; Coe et al., 2014; Collins et al., 2020). Landslides from the April 2020 storm, one of the events with the smallest $LPA$ values in our catalogue, were triggered by an isolated thunderstorm following a persistent, multi-day period of rainfall associated with a cutoff low-pressure system (CW3E, 2023). The mapped landslides spatially correlate with a roughly 10-km wide area of high (>50 dBZ) radar reflectivity representing the isolated effects of the thunderstorm, which also corresponds to a local peak in $A^*$ (Fig. 9a). In a similar example of landslide control by mesoscale processes, extreme rainfall in the March 2018 Central California/Sierra Nevada storm event was influenced by a narrow cold-frontal rainband (NCFR) that stalled over the region following the passage of an AR4 atmospheric river (Collins et al., 2020). Here the pattern of landsliding closely matches the radar reflectivity signature of the NCFR passage across the region as well as the pattern of $A^*$ (Fig. 9b). These two cases in particular highlight how synoptic and mesoscale atmospheric features may work together to produced localized landsliding. In each case, the synoptic feature (cutoff-low or atmospheric river) provided long-duration rainfall which sufficiently primed the soils for failure. This was followed by a high-intensity, short-duration burst of rainfall from a mesoscale feature that acted as a landslide trigger (e.g., Collins et al., 2020; Bogaard and Greco, 2018). The resultant footprint of $A^*$ in these two examples thus directly reflects the passage of mesoscale features.

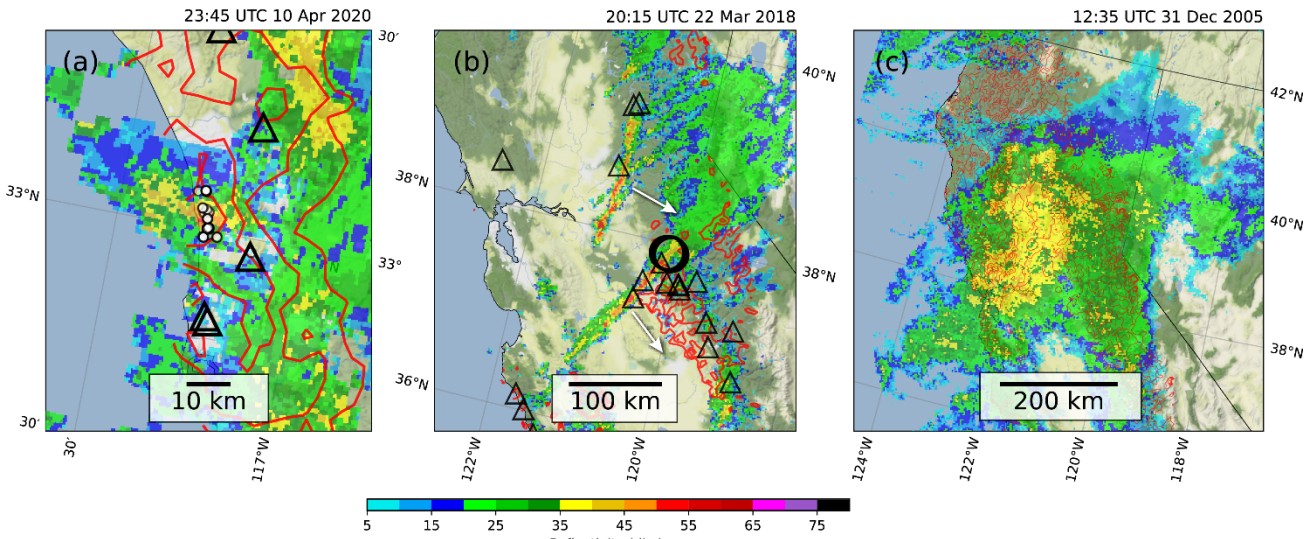

Figure 9. Maps showing examples of a range of spatial scales of precipitation influencing landslide-inducing rainfall distribution in California. Radar reflectivity values are shown as colored pixels, and associated $A^*$>1 values for each storm are shown as red contours. (a) Mesoscale features such as isolated thunderstorms produced very high intensity rainfall and led to localized landslide hotspots in the April 2020 storm in southern California (Table 1; black circles). (b) Narrow cold frontal rainbands (NCFR), on the order of a few km wide and tens of km long, are another mesoscale feature that can produce high-intensity rainfall, leading to regional zones of landsliding as was the case along the Sierra Nevada foothills during the March 2018 event (Table 1). Large circles are mapped landslides from Corbett et al. (2020), triangles are National Weather Service local storm reports of slope failures during the event (Iowa Environmental Mesonet Cow, 2023), and white arrows show the propagation direction of the NCFRs. (c) Broad areas of persistent moderate-intensity precipitation may develop under favorable atmospheric conditions, as in the December 2005 storm in northern California (Table 1), whichcan lead to widespread distributions of landslide potential area (*LPA*) when antecedent conditions are sufficiently high over widespread mountainous terrain (Fig. 7c). Map tiles copyrighted by Stamen Design, 2023, under a Creative Commons Attribution (CC BY4.0) license.

Conversely, the highest-magnitude *LPA* event in the dataset, the December 2005 storm in northern California (*LPA* = 38,600 km$^2$), was associated with persistent (multi-hour) moderate intensity rainfall over broad areas (~200 km-scale) (Fig. 9c). This may occur due to the persistence of AR conditions over an area or from increased precipitation rates associated with the development of mesoscale frontal waves or secondary cyclones developing near landfalling ARs (e.g., Martin et al. 2019), among other atmospheric processes. The observed rainfall intensities were not as high as the other two events featuring well-defined mesoscale high-intensity rainfall features, but the persistence of moderate-intensity rainfall over an area with very high antecedent $A^*$ (Fig. 7c) resulted in excessively anomalous rainfall at the large regional scale. This region of northern California also has a broader concentration of mountainous terrain than elsewhere in the state (e.g., Fig. 6), which will inherently result in a larger *LPA* given similar meteorological conditions. Taken together, these results suggest spatial patterns of multi-hour moderate intensity precipitation and short-duration, high-intensity rainfall can both impact a

storm's resulting $LPA$ depending on the antecedent $A^*$ distribution. If hillslope soils are relatively dry over a region, then the post-event pattern of $A^*$ may closely resemble the shape of the meteorological structures that yield the highest-intensity precipitation, which typically occur at a finer spatial scale (e.g., Fig 9a,b). If the soils preceding a landfalling storm are relatively wet over a broad region, then the pattern of landslide-inducing rainfall may reflect more the larger meteorologic structures that yield moderate-intensity rainfall (e.g., Fig. 9c). In this way, for many storms the distribution of antecedent $A^*$ may act as an aperture that limits what meteorologic structures imprint themselves on the landscape via distributed shallow landsliding.

Due to the role of mesoscale processes in driving landslide-inducing rainfall (Wooten et al., 2008; Minder et al., 2009; Coe et al., 2014; Collins et al., 2020), the quality of the quantitative precipitation estimates (QPEs) used in $A^*$ and the resultant $LPA$ is important. QPEs that incorporate radar observations may better capture smaller-scale convective features that may not be represented by interpolated rain gauge observations such as the CNRFC 6-hourly QPE (CNRFC, 2023). This is particularly true in landscapes where rain gauges may be heterogeneously distributed. For example, the NCFR passage that drove landsliding in the March 2018 storm was captured well by radar but not particularly well in the rain gauge-interpolated precipitation dataset along the Sierra Nevada foothills where gauge data are relatively sparse (Fig. 9b). However, QPEs incorporating radar observations may be limited by radar coverage in the complex terrain of the western United States.

**5.3 Evaluating $A^*$ performance at the statewide level: an example from the Winter 2023 atmospheric river sequence**

Concerns also remain as to the degree of predictive success for $A^*$ across a broader range of events and beyond the relatively small regions (10s to 10,000s km$^2$) used for model calibration (e.g., Fig. 3). More systematic and complete landslide inventories are therefore needed at the mega-regional scale (i.e., many 100,000 km$^2$; California is 424,000 km$^2$ in size) to better evaluate how variations in $A^*$ map with changes in both the presence and absence of landslides and their relative spatial density. Further, if these indices are utilized in decision-making schemes for evaluating risk and regional warning criteria, more work is required to not only examine a broader range of events, particularly large storms that did not trigger landslides, but to examine how $A^*$ correlates to landslide triggering across changes in parameters such as topography, lithology, vegetation, and evapotranspiration. For example, Marc et al. (2019) showed that increasing $R^*$ scales with increasing landslide spatial density, and that accounting for lithologic differences further increased correlation. The $A^*$ threshold in this analysis is designed to signify regions of widespread shallow landsliding, but to what extent do increases in $A^*$ correlate with increases in shallow landslide density, and how sensitive a predictor is $A^*$ for more isolated landslide events?

Recently, California experienced an extreme storm sequence of nine back-to-back atmospheric river arrivals from December 2022 to January 2023 (DeFlorio et al., 2024), driving statewide impacts including flooding, landslides and debris

flows, and significant wind damage that produced an estimated \$5-\$7 billion in damages (Moody's RMS, 2023). Throughout the emergency response to the ongoing impacts, the California Geological Survey (CGS) collated and verified reported landslides from State and Federal government agencies (i.e., Brien et al., 2023), social media, California Highway Patrol, news reports, and citizen submissions to include in the CGS Reported California Landslides Database (2023). The resulting inventory includes over 700 landslide reports from across the state, mostly nearby road networks where observers were located. Although the inventory does not include a full, detailed accounting of shallow landslides from satellite imagery (e.g., Fig. 3), it covers the entire California study area and thus provides an opportunity to explore how variations in $A^*$ throughout the AR sequence correlate with the location and relative densities of reported landslides. To examine how relative landslide density may correlate with $A^*$ magnitude, we sum the landslide points and spatially average $A^*$ maxima in 15 arc-minute (~20 km) bins (Fig. 10c).

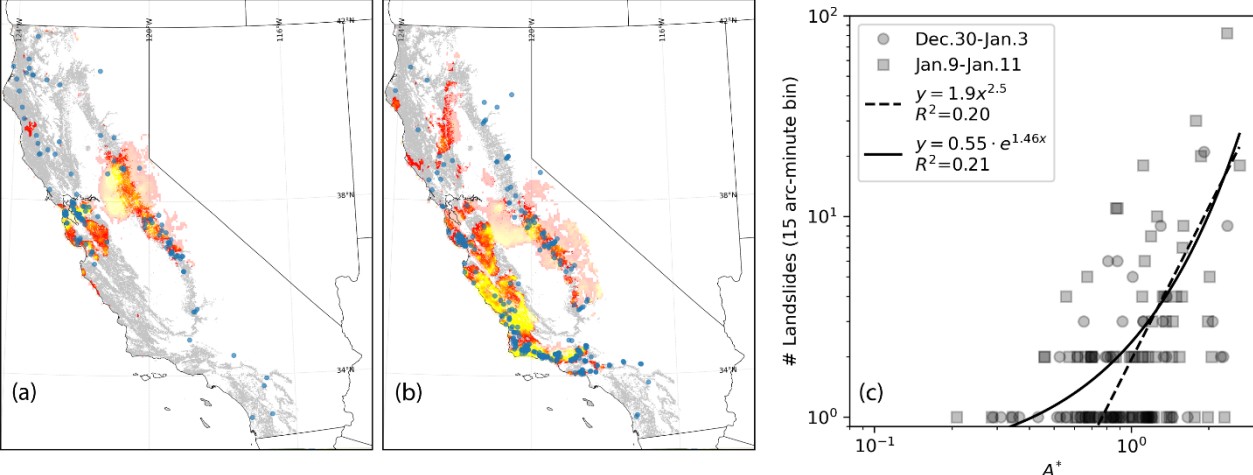

**Fig. 10. Maps showing distributions of $A^*$ maxima and reported landslides during two periods of the December 2022 - January 2023 atmospheric river sequence: (a) 30 December – 03 January which strongly impacted the San Francisco Bay area; and (b) 09 January – 11 January, which strongly impacted the central coast and southern California. Yellow symbols are landslides from each time period from the California Geological Survey Reported Landslides Database. Panel (c) shows a plot relating a grid of landslide point density (y-axis) to $A^*$ maxima for each respective period in the storm sequence. Although a number of isolated slides show low values of $A^*$, as $A^*$ approaches 1 landslide density begins to rise rapidly. This highlights the efficacy of the method for identifying zones of widespread landsliding rather than locally isolated events. Shaded relief in (a) and (b) from NASA SRTM 30 m DEM (NASA, 2013).**

Figure 10 shows snapshots of $A^*$ maxima and reported landslides from the CGS database during two of the most intense storm periods during the 2023 AR sequence: 30 December 2022 – 03 January 2023 (AR3; Fig. 10a), and 09 January – 11 January 2023 (AR3; Fig. 10b). Overall, the footprints of $A^*$ generally cover the zones of high landslide density at the regional scale for both cases. Isolated landslides are not very well-resolved by the method; however, some events in the reported landslide database may be related to land use and may not reflect purely natural conditions. Additionally, at this level of mapping it is difficult to evaluate false positives (i.e., zones of above-threshold $A^*$ where reported landslides are absent) because of potential reporting biases. For example, landslides may be under-reported in areas of low road or population density, or in instances when certain roads may have already been closed due to storm damage. Future work with a more robust mapping of natural failures across the entire domain (a time-consuming but essential effort) would help quantify prediction uncertainty and likely provide more statistically robust relationship between increasing $A^*$ and landslide spatial density. Even so, a gross comparison of reported landslide spatial density with increases in $A^*$ (Fig. 10c) shows a marked rise in landslide spatial density as $A^*$ approaches and exceeds a value of 1, our calibrated threshold based on the local 10-year recurrence of $AWI$.

Although the climatic normalization process does appear to account for some regional landslide susceptibility differences potentially driven by the geomorphic tuning of the landscape to the regional climate (e.g., Marc et al., 2019), more local site heterogeneity in soil strength and root cohesion likely exert a strong second order control on the relationship landslide triggering during extreme rainfall events (e.g., McGuire et al., 2016; Rengers et al., 2016; Peruccacci et al., 2017). The analysis presented here indicates, however, that to first order $A^*$ is an effective metric for delineating zones of widespread landsliding and can hence serve as a useful guide for evaluation of regional hazard potential.

**5.4 Towards predicting the effects of rainfall-induced landslide hazards**

A primary goal of this analysis was to work towards enhancing situational awareness for rainfall-induced shallow landslide hazard. Global forecast models such as the Global Forecast System (NOAA, 2023) and the European Center for Medium-Range Weather Forecasts (ECMWF, 2024) provide precipitation forecasts out to approximately two weeks and can be used as input to provide forecasts of $A^*$ and $LPA$. Gridded precipitation estimates such as the NOAA Stage IV product (Seo and Breidenbach, 2002; Nelson et al., 2016) could be used to calculate season-to-date $A^*$ in an operational scenario, which itself can provide a glimpse into what potential impacts from an incoming storm may look like (e.g., Fig. 7c). In data-poor regions where calibrated gridded precipitation datasets may not be available, this methodology could be tested using globally available satellite-derived precipitation products. Although the methods developed herein are only applicable for situational awareness of hazardous rainfall at the scale of the precipitation data used and which are typically coarser than the spatial scale of individual hillslopes, one potential advantage is its simplicity of implementation as only rainfall data are needed as model input for $A^*$. In future work, a more rigorous investigation of model rate constants and additional controls on the water mass balance such as evapotranspiration could be investigated. This could be particularly important for Mediterranean

climates like California, which are projected to see an increasing number of dry days in a warming climate (Polade et al., 2014).

Nevertheless, our analysis shows that $A^*$ provides a good first-order indication of landslide-inducing rainfall for soil-mantled hillslopes across a range of climatic conditions in California. This simple approach could be used with precipitation forecasts and estimates to provide early warning of landslide hazards and support emergency management decisions ahead of potential events. Additionally, the approach presented here can be used to provide insight into the meteorological and climatic processes that control landslide hazard, conduct intercomparisons of past landslide events, or be used for climate model output to assess the potential for increased landslide hazard in future storm events.

**Data Availability**

Landslide data supporting this manuscript are available as U.S. Geological Survey Data Releases (Corbett and Perkins, 2024a; 2024b).

**Disclaimer**

Any use of trade, firm, or product names is for descriptive purposes only and does not imply endorsement by the U.S. Government.

**Author Contributions**

JP designed and conducted analysis with input from NO, BC, PB, and SC. SC mapped pre-2023 landslides with input from JP, and PB compiled 2023 landslide data and helped with analysis. JP wrote the manuscript with input from all authors.

**Acknowledgements**

Samuel Bartlett (CW3E), Dianne Brien (USGS), Karimah Comstock (USGS), and Mikael Witte (Naval Postgraduate School) provided helpful discussions throughout the development of this work. Brian Kawzenuk (CW3E) provided AR scale calculations. Reviews by Matthew Thomas (USGS), Odin Marc (GET), and Ben Mirus (USGS) helped improve this manuscript.

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
