# Peer review of "Characterizing the scale of regional landslide triggering from storm hydrometeorology"

_EGUsphere, 2024_

## Referee Comment (RC1)

Review of "Characterizing the scale of regional landslide triggering from storm hydrometeorology" by Perkins et al.,

The authors present an analysis of several storms induced landslide events, most relating to atmospheric rivers phenomena in California. They retrieve rainfall from a gridded gauge product (covering >10 years, 6hr resolution, 4km spatial resolution),  and a leaky bucket model to constrain regolith moisture and derive a soil moisture anomaly, A*, relative to a 15 yr return event. They show that 15 yr return appear to be the minimal return time for causing extensive landsliding based on 4 well constrained case and then show and discuss the advantage of using a soil moisture anomaly (rather than simple rainfall anomaly)  to understand landsliding triggered by rainfall in California. The work is a nice progression from previous work arguing for the use of anomaly to study landslide event (Rainfall anomaly for Marc et al., 2019, or soil moisture anomaly for Saito and Matusyama 2012, but with a more complex methodology and rather preliminary data).
Therefore the authors' work goes provides first basis for simple, physically meainingful and regional scale indicators that could provide a basis for landslide hazard forecasting during storms.

In terms of methodology and presentation, I had reviewed a previous version of this work and a lot of my previous concerns in terms of methodology and clarity have been addressed and this version of the draft appears very clear and well thought to me. I therefore congratulate the authors, as I think the work will be a very good contribution to Esurf !
I provide below a series of minor comments where I have identified potential improvements.

Sincerely, Odin Marc

**Line By Line Comments**

**Introduction**
→ It is maybe a personal feeling but I had the impression of a small disconnect in the Introduction with the paragraph about the storm/AR categorization… Given the work is about Rainfall induced landslides and better understanding/forecasting them I thought this could rather come after the paragraph detailing the state of the art in terms of relating landslides to rainfall and soil moisture. But this is up to the authors.

→ Up to you but it may be interesting to mention the usefulness of simple leaky barrel approaches to understand the timing and conditions of landsliding in other context than California, such as monsoon induced landslides in Nepal as presented and discussed in Gabet et al., 2004, and Burrows et al., 2023

→ Last, this is optional and up to you but I personally think within the general framework of combining basic characterization of the topography (typically slope gradient) and of the forcing to understand and forecast landsliding I think the studies on seismically triggered landsliding and rainfall-triggered landsliding are quite complementary and illustrative of similar concept.
Thus Marc et al., 2017 and Tanyas and Lombardo 2019 have basically developed and validated to some extent a Landslide Potential Area (LPA this study) for EQ induced landslides , or in their term characterizing a Landslide Affected Area, based on the intersection of a minimal slope criteria and a minimal ground shaking criteria.
So this work and following may be worth to mention in intro or discussion to introduce/discuss the LPA concept.

**Figure 2** : Fig 2D has something weird with the polygon of 2d ? And the caption is missing an explanation of what these boundaries are exactly… Also you say in the caption : "**A regionally consistent threshold would plot as a horizontal line**" Do you mean an absolute, constant threshold (thus constant across the region ) ? If yes I don't find "regionally consistent" the best term… wording here could be confusing I would say maybe better to rephrase.

**Figure 4 :** This is interesting to discuss what LPA is and from what it results but does not help to assess its validity. Could you show on Fig 4 the available landslide event data ? Basically for the 4 calibration storms could you display the landslide location (in a less zoomed manner than on Figure 2) ?

Also could you compare/ correlate : LPA to the actually measured landslide affected area ? (typically the convex hull containing all or 95% of the landsliding ? See Marc et al 2017, Tanyas and Lombardo 2019).
Another question coming is whether LPA is correlated to the total landslide area ? Did you check that ?

**Fig 5 :** I am a bit skeptical about the proposition that seasonality is the main control, or at least I wonder how important are other aspect :
Is Dec 05 also extremely high  because the storms affected the north-western part of california with more extensive hillslopes above 5°…  ? Or Because A* was not just above 1 but quite greater compare to the other storms (see Fig 2E) ?
To better test your explanation about seasonality maybe you could show/check the Area with A*>1 against seasonality, independent of hillslopes. And then maybe discuss the role of the storm location relative to the topography.

Section 5.1 / Fig 6 → nice and clear, great job showing the difference between R* and A* !
However in the caption prefer : "Little impact could be **expected/anticipated** for distributed shallow landslide occurrence" (because there is not yet a prediction system based on A*).

**Fig 7 :** This is interesting and could address some of my concern of the actual comparison between A* and landsliding (when there is data) : But for this showing only dbZ from rain radar is a step back because we lose the effect of soil moisture;
The simplest would be to show both : Show the dbZ and below the A* map derived from gauges with slopes maybe ? With the landslide report in both for comparison… This would allow to make your point more clearly or to discuss the respective limits of using dbZ or A* only to track landslide hazard.

L495-500 go in this direction but would be more clear if Fig 7 would contain both : A* derives by gauge vs dbZ for localized hotspot…  This discussion goes back to the importance of the specificity of the dataset used to derive R* or A* in your case. Radar being rare, should we use gauge or satellite QPE when we don't have it ?
Recent work such as Thomas et al 2019, Ozturk et al., 2021, Marc et al., 2022  discuss the issues of advantage, limits and potential use of satellite derived precipitation estimates for assessing landsliding.

L504 : Large regional scale

L543: "a rare and comprehensive, time-consuming effort "→ Maybe rather write "a time-consuming, but essential, effort" Indeed it is still done routinely in many areas (Japan, Taiwan for example) and has been done for a fair number of case. The current sentence could suggest to some reader that it's not an essential part of future work.

Fig 8 / L545 : Nice ! However you should make Fig 8C in Semi – Log  or Log – Log ! It's clearly a non linear trend and we cannot really see the trend and the data with low landslide density.
If it is a power-law (linear in log log) having a rough estimate of the preliminary exponent (near 1 ? >1 ? <1 ?  ) could help to derive interepretation and for comparison with other/later work would be useful.
Last, on such plot (which is a conceptually similar to Fig 2A of Marc et al 2019, where R* was used not A*) one would wonder how much the scatter of Landslide Density vs A*  would be reduced by normalizing by a slope Gradient term or separating different lithologies (possibly with different regolith thickness or strength).
Your data may only allow to do it with slope, but it could be nice to check or at least mention it.

L547-548: The phrasing here is a bit ambiguous here and may merit one or two more sentence, or rephrasing.
Because in Marc et al., 2019 we hypothesize that if the landslide density correlates with R/R10 it is because the landscape has co evolved with climatic conditions (through repeated landsliding). Indeed the landscape do experience only R (the rainfall during the storm) but it's property setting its response to R could have been influenced by the previous storms, and thus correlates with R10. I think the same reasoning apply here with A and A15.
So the sentence oppose soil strength / root / vegetation and climatic normalization whereas the understanding of this normalization (as proposed in Marc et al 2019, and in some geomorphological references) is that at least some of these parameters are captured by the past extreme statistics (R10 or here A15).
So unless you put forward an alternative interpretation, I would suggest that you specify (in some way) that the nomalization probably works because it does capture some of the secondary landscape parameters that control hillslope stability. Of course they may not control it all as some parameters may evolve independently of past extreme, or on faster timescales.

L554-560: This is interesting discussion toward Forecast ! However it could be nice if to add one or two sentences towards broader views: Testing the A* methods with other data sources (such as satellite derived rainfall or weather forecast models) which could be done in other geographic contexts/areas (including data poor for the satellite).

**References** (used in the review but not in the manuscript):

Burrows, K., Marc, O., and Andermann, C.: Retrieval of Monsoon Landslide Timings With Sentinel-1 Reveals the Effects of Earthquakes and Extreme Rainfall, Geophysical Research Letters, 50, e2023GL104720, https://doi.org/10.1029/2023GL104720, 2023.

Gabet, E. J., Burbank, D. W., Putkonen, J. K., Pratt-Sitaula, B. A., and Ojha, T.: Rainfall thresholds for landsliding in the Himalayas of Nepal, Geomorphology, 63, 131–143, https://doi.org/10.1016/j.geomorph.2004.03.011, 2004.

Marc, O., Meunier, P., and Hovius, N.: Prediction of the area affected by earthquake-induced landsliding based on seismological parameters, Nat. Hazards Earth Syst. Sci., 17, 1159–1175,

https://doi.org/10.5194/nhess-17-1159-2017, 2017.

Marc, O., Oliveira, R. A. J., Gosset, M., Emberson, R., and Malet, J.-P.: Global Assessment of the Capability of Satellite Precipitation Products to Retrieve Landslide-Triggering Extreme Rainfall Events, Earth Interactions, 26, 122–138, https://doi.org/10.1175/EI-D-21-0022.1, 2022.

Ozturk, U., Saito, H., Matsushi, Y., Crisologo, I., and Schwanghart, W.: Can global rainfall estimates (satellite and reanalysis) aid landslide hindcasting?, Landslides, https://doi.org/10.1007/s10346-021-01689-3, 2021.

Tanyaş, H. and Lombardo, L.: Variation in landslide-affected area under the control of ground motion and topography, Engineering Geology, 260, 105229, https://doi.org/10.1016/j.enggeo.2019.105229, 2019.

Thomas, M. A., Collins, B. D., and Mirus, B. B.: Assessing the Feasibility of Satellite-Based Thresholds for Hydrologically Driven Landsliding, Water Resources Research, 55, 9006–9023, https://doi.org/10.1029/2019WR025577, 2019.

---

## Referee Comment (RC2)

**Summary and recommendation**

This work presents a simple yet novel method for characterizing landslide potential at the regional scale using a normalized rainfall index, A*. The concept is based in part on the idea of a characteristic recurrence interval for landslide inducing storms proposed for global application by Marc et al., (2019, 2022), but introduces a substantial advance by considering the relative soil-saturation rather than a rainfall recurrence interval using the Antecedent Wetness Index (AWI), implemented for Seattle by Godt et al. (2006) and similar to a leaky-bucket approach used for the SF Bay Area by Wilson and Weiczorek (1995). The authors demonstrate the utility of this parsimonious approach across the entire state of California, a region of the U.S, with considerable landslide hazards and risk, as well as extensive event-based inventories and rainfall data to apply and test their model. The work demonstrates a nice correlation between their index exceedance and landslide occurrence as predicted by their Landslide Potential Area (LPA) indicator based on a calibration with four events and a total of nine events for demonstrating the validity of the approach.

The work is highly relevant and will be of substantial interest to landslides researchers as well as potential applications for situational awareness and near-term planning in the decision-management arena. In particular, it presents a nice contribution in terms of moving from rainfall-only driven triggering to consider the role of the subsurface and antecedent conditions at the regional scale, albeit through a simplified bucket model. This is in fact remarkably well-aligned with the type of approaches we propose in a recent commentary that is currently open for discussion (Mirus et al, NHESS-Discussion, 2024). I was pleased to see that the authors demonstrated that their non-dimensional index A* appears to be more valuable than the IVT criteria of ARs, which is often called upon as a proxy for landslide potential. So, ultimately, this is a great proof-of-concept for going beyond storm metrics used in the atmospheric sciences and adding more landslide-relevant considerations. It will surely be a nice contribution as an approach that is likely to gain traction moving forward.

Overall, the manuscript is very well-written, and the figures and tables are clear. The motivation and background are well-supported, and the work builds upon other novel contributions in the literature dating back to early contributions from previous USGS researchers (i.e., Campbell, 1975; Wilson and Wiezcorek, 1995; Godt et al., 2006) among others. The paper is short, which is nice for readability, but as a result I found there are some critical details that are missing from the methodology and some other areas for discussion that would help readers better appreciate and repeat the work. These related to both the advances and current limits of the assumptions baked into the AWI approach as implemented as well as those presented by data availability that limits more rigorous testing. These gaps can likely be addressed through the major revisions outlined in the attached text, including line-by-line comments. A revised manuscript that addresses these points should ultimately be accepted for final publication in NHESS.

**Soil moisture and AWI as a proxy**

- It is interesting to use a soil-moisture threshold, when previous work at least for many of the SF Bay Area storms, suggests that positive porewater pressures are a more relevant metric than soil moisture levels for actual triggering since near-fully saturated soils are needed to trigger landslides (e.g., Thomas et al., *Landslides*, 2017). Furthermore, the AWI concept as presented in Godt et al. (2006) was used to modulate the Intensity Duration (ID) threshold, not as a stand-alone predictor of landslides. This is important to note, particularly given the conclusions that high-intensity rainfall and other mesoscale processes may be relevant for triggering and are not consistently captured in the A* metric. Potentially this approach could be improved even further by combining A* with short-term forecasts of precipitation intensity. For example, the AWI is now used with Quantitative Precipitation Forecasts of intensity and duration threshold to provide situational awareness for the greater Seattle area (see https://www.usgs.gov/media/images/seattle-washington-landslide-monitoring-site-awi).
- As the authors note, the AWI is designed to fill quickly since rain is added instantaneously, and it also drains more quickly than real soils (see below). So, in many ways it is more appropriate as an antecedent metric and not as a triggering metric. In this context, why would you expect this to work better than rainfall-only thresholds identified with other modeling approaches (e.g., Scheevel et al., *USGS*, 2017; Patton et al, *NHESS*, 2023)? We have some discussion about this in a paper about Seattle (Mirus et al. *Landslides*, 2018) and something similar in scope might be useful to flush this topic out a little more fully in the discussion.
- Most importantly, it appears spatial variability in the AWI calculations is entirely a result of variable rainfall across the State. Did I miss something? If not, this is a very significant simplifying assumption that needs to be stated more explicitly. I understand that the point of this paper is largely as a proof of concept for the simple index approach, and it does that successfully, but it's not clear how the technique could / should be implemented for a given site of interest. Here are a few additional points to consider:

  1. Technically the A* is not a model parameter, it's an index/proxy simulated by the model. However, the AWI does use at least one parameter, kd, which wasn't discussed. It is a lumped fitting parameter that reflects the soil drainage rate, which integrates many the soil properties (Ksat, retention curves, porosity, thickness, etc.), as well as other geometric factors that affect drainage (slope, convergence, soil thickness, etc.). Although the AWI is indented to be representative over a broad area and assumes infiltration is added instantaneously, there is no reason to assume that all the soils in California will drain at the same rate. In the revisions, please discuss why this assumption is ok enough for your objectives and how it potentially influences the results.
  2. The equation in Godt et al. (2006) accounts for ET in calculating effective rainfall Ii. Did this study also use ET? If not clarify that and explain why. Godt et al. (2006) used a fairly simple monthly estimate of PET, which was good enough for calculating antecedent conditions at the hourly scale, but has a major impact on the outcome if ignored, particularly during spring months when ET ramps up.

3. Given that the USGS has published soil moisture data for at least one of the BALT sites with some landslide-inducing storms that you considered (i.e., Thomas et al., USGS-Data, 2017), have you looked at how these compare qualitatively to your AWI calculations for those grid cells? Godt et al. (2006) developed their parameterization of the AWI based on observed soil moisture data from Edmunds, WA (refer to Figure 4 in their paper). With sufficient calibration, one can get an AWI that nicely matches observed soil moisture dynamics for a region of interest (for example, see below).

[Figure]

**Figure**: time series plot of AWI calculated using parameterization from Godt et al. (2006) as displayed on the USGS website (https://www.usgs.gov/media/images/seattle-washington-landslide-monitoring-site-awi) versus mean observed saturation from in-situ soil moisture sensors in Mukilteo, WA (refer to Mirus et al., *WRR*, 2017). The AWI wets up more quickly at the beginning of the rainy season, then matches observations for the higher saturated values reasonably well, but not as well during the dry season as the soil drains below field capacity and approaches wilting point. Whereas at the event scale the AWI drains too slowly relative to real soils, at the seasonal scale it drains too quickly.

**Data limitations and false/failed alarms**

- The period of record with only 15-years does not adequately support the conclusion of a 15-year recurrence interval for identifying A* and suggests a longer record is needed to better constrain this recurrence-interval based ratio. I didn't quite understand why the full 19 years of the rainfall dataset wasn't leveraged fully. Additionally, although the input rainfall data is time limited, one could easily calculate the AWI going further back with

different time-series of rainfall databases to get a sense of whether 15 years is close enough or if much longer records are needed to capture this relevant recurrence interval.

- Any criteria used for landslide assessments, whether temporal, spatial, or spatiotemporal should in some way consider the potential for false or failed alarms. I understand that there are data limitations that likely preclude a rigorous quantitative analysis of performance or even explicit consideration of failed alarms or false alarms, but some discussion is needed. Can you identify at least a few "big" storms that might have triggered landslides and did not? Were they also associated with positive (or high) LPA values? Alternatively, from the storms you did evaluate are there any substantial areas with some LPA that *didn't* produce any landslides (probably, given your very conservative 5-degree slope threshold), and if so, how much?

- Can you correlate the LPA to landslide densities, numbers of landslides, or even just a general qualitative descriptor of how widespread landslides were within the areas predicted? Does a higher LPA link to a greater number/severity of landsliding? I ask this because the magnitude varies so widely and the slope is so low, that it seems some guidance would be needed to understand what the difference might be between LPA = 60 vs. LPA = 11k. Given the conservative slope threshold, I could easily imagine a scenario in which lower LPA in a localized area of very steep terrain triggers a higher density of landslides than a broader area of less steep terrain with a much higher total LPA.

**Specific line-by-line comments:**

12 – Can you provide a few words that better characterize A* for the abstract? Is it a proxy, or a dimensionless index that you test as a proxy for landslide probability? It's not clear until I read the paper.

54 – Consider mentioning however, that most of those ARs from their analysis did NOT result in landslide triggering. So, there's more going on than just high IVT and ARs.

87 – I'm not sure that this Bogaard and Greco paper is an appropriate citation for the ID threshold since it largely focuses on the inconstancies and advocates for moving away from the approach.

115 – It's worth citing some of the other recent literature using direct measurements of soil water content to identify empirical thresholds (see Mirus et al., NHESS Discussion, 2024 and references therein).

156 – Technically this isn't a parameter, but rather an index or nondimensional ratio that is the output of your calibrated/parameterized AWI model.

178 – Godt et al. (2006) used an effective rainfall minus PET to calculate Ii, so that's an important consideration especially for California, particularly to consider potential impacts of climate change.

195 – Consider the Thomas et al., (*WRR,* 2019) not only showed that SMAP over-smoothed the soil moisture data, but PRISM also over-smoothed out the orographic effect on precipitation intensity and totals. Your dimensionless approach with A* should help avoid this issue of magnitude of precipitation (or soil moisture) measurements, which is a common issue with satellite precipitation estimates.

203 – Again, 15-years seems like a short window considering your conclusion and that Marc et al. (2022) found that the 10-year rainfall anomaly was potentially only 50% effective in characterizing landslide potential for extreme and widespread landslide events. Some discussion or further analysis using a proxy with older rainfall records for one (or more) location seems prudent to better support your conclusion.

207 – Technically you're not calibrating A* since there isn't any parameter adjustment, you're identifying which value of A* corresponds to landslide occurrence in the same way one might use a critical factor of safety for estimating slope failures.

Also, refer to general comment about kd. Was the soil variability considered in the field capacity, etc. considered for calibrating kd in area without monitoring data or landslide occurrence data?

223 – Consider specifying how the interpolation is applied (linear? Krigging? Other?) rather than citing a toolbox and assuming everyone knows how it works.

225 – Why not use a 19-year time series to identify the anomaly?

236 – Why not use a more sophisticated approach than 5-degree slope? For example, the Godt et al., *NASL*, 2012) model is available at 1km resolution and Brabb et al, *USGS,* 1999 found a 25-degree slope was suitable for rapid shallow landslides and debris flows. Within this context your threshold is very conservative. Discuss?

Figure 2. This relationship does not appear to be 1:1 and suggests that your AWI might need to vary for different landslide triggering storms or locations (see general comment #2 about calibrating kd). How do your other 5 storms plot here?

423 – Do you mean Luna and Korup, 2022?

247 – What about false alarms?

266-67 – I don't quite follow this. Isn't the maximum AR value used going to be independent of location to produce the value and time of AR max? How does it correspond to the location of one or more observed landslides?

Table 1 – This includes results and should probably be linked to those later in the paper. Also, can you list the number or density of landslides you showed in Figure 8c (albeit, maybe acknowledging that the inventories are incomplete)?

What about non-events? Are you able to introduce any of these, even anecdotally?

Figure 4 – This is a European/international journal, so I suggest you clarify that the numbers are years (e.g., January 2005, not January 5$^{th}$)

453 – Again, a good place to point out that Cordeira found that while most landslides were associated with ARs, most ARs didn't necessarily produce landslides.

465 – Maybe I missed it, but your dataset considered 6h rainfall, so how does that account for the high-intensity bursts?

Fig 8c. This is the most compelling argument for the broad utility of this approach. You might be able to do even better if your kd values varied by soil type and ET were considered in calculation of AWI.

540 – Here or elsewhere warrants some discussion of false positives and false negatives, which are not tested thoroughly. Even if this is justified by a lack of comprehensive data, this is still a limitation of the approach worth discussing (if only to point out the data needs for further developing this method!).

548 – Again, what about local variation in soil properties (depth, ksat, retention curve, etc.) all of which are wrapped up in the kd parameter that you seem to have held constant for the entire State of California?!

569 – This potential value underscores the need to consider ET in the effective rainfall Ii.

---

## Author Comment (AC1)

**Response to reviewer Marc comments**

Note: As in our response to Reviewer Mirus, here the Reviewer text is in black, and our responses are in blue.

**Marc Review of "Characterizing the scale of regional landslide triggering from storm hydrometeorology" by Perkins et al.,**

The authors present an analysis of several storms induced landslide events, most relating to atmospheric rivers phenomena in California. They retrieve rainfall from a gridded gauge product (covering >10 years, 6hr resolution, 4km spatial resolution), and a leaky bucket model to constrain regolith moisture and derive a soil moisture anomaly, A*, relative to a 15 yr return event. They show that 15 yr return appear to be the minimal return time for causing extensive landsliding based on 4 well constrained case and then show and discuss the advantage of using a soil moisture anomaly (rather than simple rainfall anomaly) to understand landsliding triggered by rainfall in California. The work is a nice progression from previous work arguing for the use of anomaly to study landslide event (Rainfall anomaly for Marc et al., 2019, or soil moisture anomaly for Saito and Matusyama 2012, but with a more complex methodology and rather preliminary data).Therefore the authors' work goes provides first basis for simple, physically meaningful and regional scale indicators that could provide a basis for landslide hazard forecasting during storms.

In terms of methodology and presentation, I had reviewed a previous version of this work and a lot of my previous concerns in terms of methodology and clarity have been addressed and this version of the draft appears very clear and well thought to me. I therefore congratulate the authors, as I think the work will be a very good contribution to Esurf! I provide below a series of minor comments where I have identified potential improvements.

Sincerely, Odin Marc

We thank Reviewer Marc for providing some extremely helpful comments that led us to think more about what the data are indicating in terms of the processes at play during the storms in our catalog, and to ultimately produce a much stronger and more considered manuscript.

**Line By Line Comments**

**Introduction**
→ It is maybe a personal feeling but I had the impression of a small disconnect in the Introduction with the paragraph about the storm/AR categorization… Given the work is about Rainfall induced landslides and better understanding/forecasting them I thought this could rather come after the paragraph detailing the state of the art in terms of relating landslides to rainfall and soil moisture. But this is up to the authors.

Thank you for the suggestion. We agree that repositioning the paragraphs here will indeed lead to a more logical flow of ideas and draw focus to the primary aims of the manuscript.

→ Up to you but it may be interesting to mention the usefulness of simple leaky barrel approaches to understand the timing and conditions of landsliding in other context than California, such as monsoon induced landslides in Nepal as presented and discussed in Gabet et al., 2004, and Burrows et al., 2023.

Thank you for the suggestion, we will be sure to fold in these additional references.

→ Last, this is optional and up to you but I personally think within the general framework of combining basic characterization of the topography (typically slope gradient) and of the forcing to understand and forecast landsliding I think the studies on seismically triggered landsliding and rainfalltriggered landsliding are quite complementary and illustrative of similar concept.
Thus Marc et al., 2017 and Tanyas and Lombardo 2019 have basically developed and validated to some extent a Landslide Potential Area (LPA this study) for EQ induced landslides, or in their term characterizing a Landslide Affected Area, based on the intersection of a minimal slope criteria and a minimal ground shaking criteria. So this work and following may be worth to mention in intro or discussion to introduce/discuss the LPA concept.

These earthquake-induced landslide studies are indeed a great parallel for this work, and in our revised text we will be sure to describe in more depth the idea of a Landslide Affected Area and its underpinnings in the literature on coseismic landslides.

**Figure 2** : Fig 2D has something weird with the polygon of 2d ? And the caption is missing an explanation of what these boundaries are exactly…

Apologies for this oversight in not specifying the black line in the caption. This line represents the GPS tracks from the post-event landslide field verification survey. This was an attempt to more honestly convey that the landslides were mapped primarily in the field rather than within a defined box using satellite/aerial imagery. In a revised manuscript we will be sure to specify this information within the figure caption.

Also you say in the caption :  **"A regionally consistent threshold would plot as a horizontal line"** Do you mean an absolute, constant threshold (thus constant across the region ) ? If yes I don't find "regionally consistent" the best term… wording here could be confusing I would say maybe better to rephrase.

Yes, in hindsight this wording is indeed confusing! Perhaps a better term would be "a universally constant AWI threshold," which we will adjust in our revision.

**Figure 4 :** This is interesting to discuss what LPA is and from what it results but does not help to assess its validity. Could you show on Fig 4 the available landslide event data ? Basically for the 4 calibration storms could you display the landslide location (in a less zoomed manner than on Figure 2)? Also could you compare/correlate LPA to the actually measured landslide affected area ? (typically the convex hull containing all or 95% of the landsliding? See Marc et al 2017, Tanyas and Lombardo 2019). Another question coming is whether LPA is correlated to the total landslide area? Did you check that?

In our revision we will be sure to place the landslide calibration sites for the relevant events in Figure 4. Unfortunately, in the case of our calibration events a comparison between the mapped landslide area and the LPA is somewhat irrelevant because the mapped calibration sites are so small spatially (10s of km or less) compared to the geography of California (>1000 km north to south).

However, we do show a more visual representation of this approach in Figure 8. Here we utilize the California Geological Survey Reported Landslides database, which underrepresents the total number of landslides but has good spatial coverage across the state, to compare to the distribution of $A^*$ for two large atmospheric river storms that occurred in January 2023. For each storm, the distribution of above-threshold $A^*$ overlaps with the zones of relatively high landslide concentration. Because landslides for each storm occur across multiple mountain ranges separated by large valleys, a convex hull around the entire inventory will over-represent the actual landslide-affected area compared to LPA; however, one could certainly attempt to do this for each specific mountain range. Further work testing this metric using spatially extensive landslide inventories may allow for a better comparison between the actual landslide-affected area and LPA.

**Fig 5 :** I am a bit skeptical about the proposition that seasonality is the main control, or at least I wonder how important are other aspect : Is Dec 05 also extremely high because the storms affected the north-western part of california with more extensive hillslopes above 5°… ? Or Because A* was not just above 1 but quite greater compare to the other storms (see Fig 2E) ? To better test your explanation about seasonality maybe you could show/check the Area with A*>1 against seasonality, independent of hillslopes. And then maybe discuss the role of the storm location relative to the topography.

We appreciate this insightful comment! Yes, where the storm track passes is certainly relevant to the resultant LPA. The northern California Coast Range has a very high distribution of slopes > 5 degrees compared to other regions of the state, so above-threshold hydroclimatic conditions should absolutely yield a higher LPA than, say, the San Francisco Bay Area and adjacent Central Valley region that contain a higher percentage of flat slopes.

As we discuss in our response to reviewer Mirus, we unfortunately discovered an error in the code that resulted in constant amount additional water added to the AWI model (0.18 m, an

equivalent of the $R$o value used in the analysis). This resulted in an artificially high threshold recurrence value of 15 years. With the corrected AWI values, the threshold recurrence interval now appears to be 10 years (see figures in our other Response document), more in line with the $R^*_{48}$ values utilized by Marc et al. (2019). We note that this change does not appear to impact the accuracy of the results, but merely shifts the threshold recurrence interval down approximately five years. The figures presented throughout this Response are now corrected to account for this error.

From examination of our revised Fig. 5 below, one can see that this Dec 2005 event does have the second highest median over-threshold $A^*$ (and highest excluding the Jan 2005 storm that is a clear outlier from the other events). So, it appears that median $A^*$ does play a role in this case, although most events are quite similar given their respective interquartile ranges. The fact that this historic storm occurred over a broad, mountainous region also played a strong role in the resultant LPA.

Regarding the role of seasonality (e.g., Fig. R1b below), we conducted a brief test to see how the antecedent $A^*$ (presumably driven by recent storm history preceding the landslide-inducing storm of interest) played a role in the resultant LPA for each storm event.

[Figure]

*Figure R1. Relationships between LPA, median over-threshold $A^*$, seasonality, and antecedent conditions for each event in our catalog. (a) shows the median over-threshold $A^*$ (with our threshold equal 1) and corresponding event LPA. Dots are colored by the AR scale of the associated storm. Panel (b) shows monthly values of LPA on the left axis, and the right axis shows the avg. monthly number of AR arrivals along the US West Coast (Mundhenk et al., 2016). Panel (c) illustrates the relationship between the event LPA (y-axis), and the median $A^*$ value of over-threshold pixels at the onset of the storm window.*

Fig. R1c shows that the largest events in the catalog have a relatively high antecedent $A^*$ condition for the grid cells that ultimately exceed $A^*=1$ during the storm, whereas the smaller-LPA events tend to have lower antecedent $A^*$ values. These data therefore indicate that antecedent $A^*$ is likely a necessary but not sufficient condition for generating a large landslide potential area. Apart from the 2005 La Conchita event (yellow circle), much of the median $A^*$ data are lumped together and do not show a strong variation with LPA, and therefore it is difficult to discern the role of storm strength through this methodology. This, and additional factors such as the storm trajectory relative to topography discussed earlier in our response) also

play a role, and we will be sure to include an enhanced discussion of these factors in a revised manuscript.

Section 5.1 / Fig 6 → nice and clear, great job showing the difference between R* and A* ! However in the caption prefer : "Little impact could be **expected/anticipated** for distributed shallow landslide occurrence" (because there is not yet a prediction system based on A*).

We can be sure to change this language in a revised manuscript.

**Fig 7:** This is interesting and could address some of my concern of the actual comparison between A* and landsliding (when there is data): But for this showing only dbZ from rain radar is a step back because we lose the effect of soil moisture; The simplest would be to show both: Show the dbZ and below the A* map derived from gauges with slopes maybe? With the landslide report in both for comparison… This would allow to make your point more clearly or to discuss the respective limits of using dbZ or A* only to track landslide hazard.

L495-500 go in this direction but would be more clear if Fig 7 would contain both: A* derives by gauge vs dbZ for localized hotspot… This discussion goes back to the importance of the specificity of the dataset used to derive R* or A* in your case. Radar being rare, should we use gauge or satellite QPE when we don't have it? Recent work such as Thomas et al 2019, Ozturk et al., 2021, Marc et al., 2022 discuss the issues of advantage, limits and potential use of satellite derived precipitation estimates for assessing landsliding.

With this figure, our primary intent was to show that rainfall characteristics at different meteorological dynamic scales can contribute to observed patterns of landslides, rather than use the radar itself as a predictive tool. However, in the case of Fig. 7b we do describe a situation where the gauge-based QPE inadequately predicts the rainfall that the radar data shows occurred at the landslide locations (Lines 495-499 in the pre-print). Given the lack of strong radar coverage in mountainous regions in California, and the variable nature there of z-R relationships needed to successfully convert radar dbZ to true rainfall intensities, we opted for a gauge-interpolated product produced by the US National Oceanic and Atmospheric Administration (NOAA). However, it is imperfect and perhaps a combined product may help coverage in areas where gauge data may be too sparse to capture rainfall features at the micro or mesoscale. In our revision we will include corresponding maps of $A^*$ alongside the radar imagery.

L504: Large regional scale

Noted. We will be sure to adjust the language appropriately here.

L543: "a rare and comprehensive, time-consuming effort "→ Maybe rather write "a time-consuming, but essential, effort" Indeed it is still done routinely in many areas (Japan, Taiwan for example) and has been done for a fair number of cases. The current sentence could suggest to some reader that it's not an essential part of future work.

This is a fair point, and we completely agree that the work of gathering comprehensive landslide location data is essential for many facets of landslide science.

Fig 8 / L545 : Nice ! However you should make Fig 8C in Semi – Log  or Log – Log ! It's clearly a non linear trend and we cannot really see the trend and the data with low landslide density. If it is a power-law (linear in log log) having a rough estimate of the preliminary exponent (near 1 ? >1 ? <1 ?  ) could help to derive interpretation and for comparison with other/later work would be useful.

The dataset used for this plot is imperfect, as the "reported landslides" from the database are not mapped comprehensively in the way that is typically done through imagery analysis. For example, although only dozens of landslides are reported in the eastern San Francisco Bay Area for the 30 Dec. 2022 storm (Fig. 8a), author Perkins observed many hundreds of landslides from a fixed wing aerial survey in this region immediately following the event. Nevertheless, we agree that it is a worthwhile endeavor to highlight the nonlinear response of landslide density to increases in $A^*$. To that end, below we've provided an example of a figure that could be incorporated in a revised manuscript, where a best-fit power law and exponential function are shown with the landslide density data. The fits are not very good, at least partially as a result of the incomplete landslide reporting that may leave many areas of high $A^*$ without correspondingly high landslide spatial densities (i.e., bottom-right corner of the plot below). Here the methodology for calculating landslide density is slightly different from what is presented in the manuscript, as we wanted to calculate actual landslide counts within a grid (a suggestion by Reviewer Mirus) rather than relative increases showcased by the kernel density approach in the pre-print (Fig. 8c).

[Figure]

*Figure R2. A plot of landslide spatial density as a function of the mean underlying $A^*$ for 15 arc-minute (~22 km at 37° latitude) bins for both storms in Fig. 8 of the manuscript. Here the data are plotted in log-log space, illustrating the nonlinear increase of landslide density with increases in $A^*$.*

Last, on such plot (which is a conceptually similar to Fig 2A of Marc et al 2019, where R* was used not A*) one would wonder how much the scatter of Landslide Density vs A* would be reduced by normalizing by a slope Gradient term or separating different lithologies (possibly with different regolith thickness or strength). Your data may only allow to do it with slope, but it could be nice to check or at least mention it.

We agree with Reviewer Marc's comments here on the role of additional factors such as topographic slope that may influence the likelihood of failure. However, given the highly coarse nature of the dataset used here, we opt to leave the data as-is and intend to explore the impact of slope on landslide spatial density in a future study where landslides can be more completely mapped across a broad region.

L547-548: The phrasing here is a bit ambiguous here and may merit one or two more sentence, or rephrasing. Because in Marc et al., 2019 we hypothesize that if the landslide density correlates with R/R10 it is because the landscape has co evolved with climatic conditions (through repeated landsliding). Indeed the landscape do experience only R (the rainfall during the storm) but it's property setting its response to R could have been influenced by the previous storms, and thus correlates with R10. I think the same reasoning apply here with A and A15. So the sentence oppose soil strength / root / vegetation and climatic normalization whereas the understanding of this normalization (as proposed in Marc et al 2019, and in some geomorphological references) is

that at least some of these parameters are captured by the past extreme statistics (R10 or here A15).

So unless you put forward an alternative interpretation, I would suggest that you specify (in some way) that the normalization probably works because it does capture some of the secondary landscape parameters that control hillslope stability. Of course they may not control it all as some parameters may evolve independently of past extreme, or on faster timescales.

We agree that adding a little more language on the nuances of what may be captured in the normalization process is worthwhile to include in our revised manuscript. Although we describe these factors more in-depth earlier on in the manuscript (Lines 126-135), we are notably lacking here in the discussion when circling back to these factors and what parameters governing slope stability may be wrapped up in $A^*$.

L554-560: This is interesting discussion toward Forecast ! However it could be nice if to add one or two sentences towards broader views: Testing the A* methods with other data sources (such as satellite derived rainfall or weather forecast models) which could be done in other geographic contexts/areas (including data poor for the satellite).

Thank you for this suggestion. We will certainly bring in a bit more discussion on how to test this approach using different datasets, particularly as we are using a QPE product that is limited to the states of California and Nevada and therefore a more general product will need to be tested using other data.

**References** (used in the review but not in the manuscript):

Burrows, K., Marc, O., and Andermann, C.: Retrieval of Monsoon Landslide Timings With Sentinel-1 Reveals the Effects of Earthquakes and Extreme Rainfall, Geophysical Research Letters, 50, e2023GL104720, https://doi.org/10.1029/2023GL104720, 2023.

Gabet, E. J., Burbank, D. W., Putkonen, J. K., Pratt-Sitaula, B. A., and Ojha, T.: Rainfall thresholds for landsliding in the Himalayas of Nepal, Geomorphology, 63, 131–143, https://doi.org/10.1016/j.geomorph.2004.03.011, 2004.

Marc, O., Meunier, P., and Hovius, N.: Prediction of the area affected by earthquake-induced landsliding based on seismological parameters, Nat. Hazards Earth Syst. Sci., 17, 1159–1175, https://doi.org/10.5194/nhess-17-1159-2017, 2017.

Marc, O., Oliveira, R. A. J., Gosset, M., Emberson, R., and Malet, J.-P.: Global Assessment of the Capability of Satellite Precipitation Products to Retrieve Landslide-Triggering Extreme Rainfall Events, Earth Interactions, 26, 122–138, https://doi.org/10.1175/EI-D-21-0022.1, 2022.

Ozturk, U., Saito, H., Matsushi, Y., Crisologo, I., and Schwanghart, W.: Can global rainfall estimates (satellite and reanalysis) aid landslide hindcasting?, Landslides, https://doi.org/10.1007/s10346-02101689-3, 2021.

Tanyaş, H. and Lombardo, L.: Variation in landslide-affected area under the control of ground motion and topography, Engineering Geology, 260, 105229, https://doi.org/10.1016/j.enggeo.2019.105229, 2019.

Thomas, M. A., Collins, B. D., and Mirus, B. B.: Assessing the Feasibility of Satellite-Based Thresholds for Hydrologically Driven Landsliding, Water Resources Research, 55, 9006–9023, https://doi.org/10.1029/2019WR025577, 2019.

---

## Author Comment (AC2)

**Response to reviewer Mirus comments**

**Summary and recommendation**

This work presents a simple yet novel method for characterizing landslide potential at the regional scale using a normalized rainfall index, A*. The concept is based in part on the idea of a characteristic recurrence interval for landslide inducing storms proposed for global application by Marc et al., (2019, 2022), but introduces a substantial advance by considering the relative soil-saturation rather than a rainfall recurrence interval using the Antecedent Wetness Index (AWI), implemented for Seattle by Godt et al. (2006) and similar to a leaky-bucket approach used for the SF Bay Area by Wilson and Weiczorek (1995). The authors demonstrate the utility of this parsimonious approach across the entire state of California, a region of the U.S, with considerable landslide hazards and risk, as well as extensive event-based inventories and rainfall data to apply and test their model. The work demonstrates a nice correlation between their index exceedance and landslide occurrence as predicted by their Landslide Potential Area (LPA) indicator based on a calibration with four events and a total of nine events for demonstrating the validity of the approach.

The work is highly relevant and will be of substantial interest to landslides researchers as well as potential applications for situational awareness and near-term planning in the decision management arena. In particular, it presents a nice contribution in terms of moving from rainfall-only driven triggering to consider the role of the subsurface and antecedent conditions at the regional scale, albeit through a simplified bucket model. This is in fact remarkably well-aligned with the type of approaches we propose in a recent commentary that is currently open for discussion (Mirus et al, NHESS-Discussion, 2024). I was pleased to see that the authors demonstrated that their non-dimensional index A* appears to be more valuable than the IVT criteria of ARs, which is often called upon as a proxy for landslide potential. So, ultimately, this is a great proof-of-concept for going beyond storm metrics used in the atmospheric sciences and adding more landslide-relevant considerations. It will surely be a nice contribution as an approach that is likely to gain traction moving forward.

Overall, the manuscript is very well-written, and the figures and tables are clear. The motivation and background are well-supported, and the work builds upon other novel contributions in the literature dating back to early contributions from previous USGS researchers (i.e., Campbell, 1975; Wilson and Wiezcorek, 1995; Godt et al., 2006) among others. The paper is short, which is nice for readability, but as a result I found there are some critical details that are missing from the methodology and some other areas for discussion that would help readers better appreciate and repeat the work. These related to both the advances and current limits of the assumptions baked into the AWI approach as implemented as well as those presented by data availability that limits more rigorous testing. These gaps can likely be addressed through the major revisions outlined in the attached text, including line-by-line comments. A revised manuscript that addresses these points should ultimately be accepted for final publication in NHESS.

We thank Reviewer Mirus for his thorough and insightful comments that challenged us to consider all aspects of our modeling framework, particularly the assumptions that go into our universal parameterization of our hydrologic tank model. As in our response to Reviewer Marc, the original reviewer comments are in black and our responses are in blue text.

**Soil moisture and AWI as a proxy**

- It is interesting to use a soil-moisture threshold, when previous work at least for many of the SF Bay Area storms, suggests that positive porewater pressures are a more relevant metric than soil moisture levels for actual triggering since near-fully saturated soils are needed to trigger landslides (e.g., Thomas et al., *Landslides*, 2017). Furthermore, the AWI concept as presented in Godt et al. (2006) was used to modulate the Intensity Duration (ID) threshold, not as a stand-alone predictor of landslides. This is important to note, particularly given the conclusions that high-intensity rainfall and other mesoscale processes may be relevant for triggering and are not consistently captured in the A* metric. Potentially this approach could be improved even further by combining A* with short-term forecasts of precipitation intensity. For example, the AWI is now used with Quantitative Precipitation Forecasts of intensity and duration threshold to provide situational awareness for the greater Seattle area (see https://www.usgs.gov/media/images/seattle-washington-landslide-monitoring-site-awi).

The hydrologic tank model presented as the antecedent water index AWI in Godt et al. (2006) was initially introduced by Wilson and Wieczorek (1995) as a proxy for pore water pressure at a USGS landslide monitoring site in La Honda, CA. Here the model was calibrated to local piezometer records and shown to have reasonably good agreement (e.g., their Fig. 9), thus it was able to capture pore water pressure evolution from local high-intensity hourly rainfall bursts as well as drainage between rainfall pulses.

This class of leaky-barrel models essentially represents a conceptual 1-D mass balance of water entering and exiting a soil column. Thus, to the extent that a soil column may experience the growth and decay of a water table throughout a storm sequence after certain antecedent conditions are met, the piezometric and soil moisture records may appear similar, and this is likely why the leaky-barrel/AWI model has been used successfully both in the context of soil moisture and pore pressure evolution during storms. The fact that the same essential model is utilized for capturing separate hydrologic variables (pore water pressure in the case of Wilson and Wieczorek, 1995, and soil moisture content in Godt et al., 2006) indeed causes some confusion, and this is perhaps made worse by the fact that we refer to the model as AWI in our manuscript while remaining agnostic as to which process the model specifically represents and instead consider it as a general metric for the mass balance of water within a grid cell. In our revised manuscript we will be sure to clarify that we are not intending to specifically capture soil moisture content but soil water more generally (which could be reflected in both moisture content and pore water pressure depending on the presence of a piezometric surface and to what degree the vadose zone is vertically equilibrated).

As the leaky barrel model takes rainfall data as input, it immediately responds to high-intensity pulses of rainfall once antecedent conditions are met. This is one of the primary reasons we chose to use a leaky barrel model, as it responds to both immediate, short-wavelength rainfall (first term in our eqn. 2) and long-term filling and draining (second term in our eqn. 2). Thus, it is a single variable that captures both rainfall intensity and duration effects, and in fact Wilson and Wieczorek (1995) show that the AWI/leaky-barrel model can be used to reconstruct rainfall intensity-duration curves (their Fig. 14). Although Godt et al. (2006) only utilize AWI as a threshold in their decision tree to then pivot to locally calibrated rainfall-intensity-duration thresholds during the passage of the actual storm, if one relaxes their assumptions about what hydrologic process AWI represents (soil moisture vs. pore water pressure) and instead considers it an integrated measure of soil water that reflects both rainfall intensity and duration, there is likely no real need to pivot to rainfall I-D thresholds and Godt et al. (2006) could have let the AWI model continue past the pre-storm period as the model originators (Wilson and Wieczorek, 1995) intended.

- As the authors note, the AWI is designed to fill quickly since rain is added instantaneously, and it also drains more quickly than real soils (see below). So, in many ways it is more appropriate as an antecedent metric and not as a triggering metric. In this context, why would you expect this to work better than rainfall-only thresholds identified with other modeling approaches (e.g., Scheevel et al., *USGS*, 2017; Patton et al, *NHESS*, 2023)? We have some discussion about this in a paper about Seattle (Mirus et al. *Landslides*, 2018) and something similar in scope might be useful to flush this topic out a little more fully in the discussion.

[Figure]

**Figure**: time series plot of AWI calculated using parameterization from Godt et al. (2006) as displayed on the USGS website (https://www.usgs.gov/media/images/seattle-washingtonlandslide-monitoring-site-awi) versus mean observed saturation from in-situ soil moisture sensors in Mukilteo, WA (refer to Mirus et al., *WRR*, 2017). The AWI wets up more quickly at the beginning of the rainy season, then matches observations for the higher saturated values reasonably well, but not as well during the dry season as the soil drains below field capacity and approaches wilting point. Whereas at the event scale the AWI drains too slowly relative to real soils, at the seasonal scale it drains too quickly.

We appreciate the concern about AWI perhaps not capturing the drainage rates of soil accurately. However, in the example supplied above the drainage rates appear to be well-captured by AWI during the wet season (the absolute values should not scale directly with soil saturation but may serve as a proxy for relative change), and the primary discrepancy is during the dry-season drainage. Here it is very important to note that the AWI model by design only drains back to a value of 0 (e.g., eqn. 2 in our pre-print). Any reduction in AWI below zero (orange dashed line in above figure) and back towards its residual value can only be achieved by another process such as evapotranspiration that is added externally (which is implemented in Godt et al., 2006). So in the example case provided by Reviewer Mirus, the primary discrepancy between AWI and soil saturation is likely due to a high rate of E-T applied to the model that results in a more rapid drawdown of summertime AWI once the drainage process ends at AWI=0.

With regard to utilizing a leaky barrel (or tank) hydrologic model to as a triggering metric, we appeal to our response to the previous comment where we cite the original intention of the AWI model as a proxy for destabilizing pore-water pressure in Wilson and Wieczorek (1995).

- Most importantly, it appears spatial variability in the AWI calculations is entirely a result of variable rainfall across the State. Did I miss something? If not, this is a very significant simplifying assumption that needs to be stated more explicitly. I understand that the point of this paper is largely as a proof of concept for the simple index approach, and it does

that successfully, but it's not clear how the technique could / should be implemented for a given site of interest. Here are a few additional points to consider:

In preparing answers to both reviewer comments, first-author Perkins discovered an error in the code to calculate $A^*$ from AWI, where a value of $R_o$ (equal to 0.18 in our manuscript) was added to the time series before the recurrence interval calculation described below. This had the effect of shifting the threshold recurrence interval for AWI values to 15 years. Removing this extraneous term reduces the threshold recurrence interval to 10 years, similar to the $R^*_{48}$ parameter of Marc et al. (2019). Below is a modified Fig. 2e from our manuscript showing the corrected AWI data for our four calibration landslide inventories:

[Figure]

*Figure R1. Corrected AWI values plotted against their 10-year recurrence value for our four calibration inventories used in the pre-print. As in the paper, here the 1:1 line is shown as a bold black line (x and y axes are not to scale here to help visualize all the data). The upper dashed line is 110% of the 10-year recurrence value, and the lower dashed line is the 90% value. An $A^*$ threshold of 1 here yields a true positive rate of 0.98. Author Perkins apologizes for the error, and additional figures will reflect the corrected values.*

While the spatial variability in AWI reflects only differences in rainfall across the state, the variability in $A^*$, which is here equal to AWI divided by its 10-year recurrence value at each pixel, reflects local variability in the hydroclimatology across the state. Below is a figure showing this process. Panel (a) shows the climatology of AWI from WY2004-WY2022 (cyan) from the six-hourly rainfall data in (b). The blue circles in (a) highlight the annual AWI maxima that are then used in a block maxima approach to calculate a generalized extreme value (GEV) distribution from which a recurrence interval can be estimated (c). Local AWI recurrence estimates are unique to each pixel and are thus used to calculate a time series of $A^*$.

[Figure]

This normalization therefore accounts for some amount of climate- (and extreme-precipitation-) dependent parameters such as soil thickness/strength/slope, vegetative root reinforcement, etc. (See our response to comments by Reviewer Marc for more information regarding this aspect of the manuscript). In Fig. 2e, we show that the triggering AWI for each of our four landslide inventories varies as a function of local climatology, which argues against using a constant-AWI threshold across the state (which would plot as a horizontal line). Thus, individual calibration of AWI to a given site should not matter in this framework as long as the triggering conditions reflect the local extreme value statistics of AWI over a sufficiently long climatology (here we used the 20-year climatology that was consistently available with state-wide coverage).

1. Technically the A* is not a model parameter, it's an index/proxy simulated by the model. However, the AWI does use at least one parameter, kd, which wasn't discussed. It is a lumped fitting parameter that reflects the soil drainage rate, which integrates many the soil properties (Ksat, retention curves, porosity, thickness, etc.), as well as other geometric factors that affect drainage (slope, convergence, soil thickness, etc.). Although the AWI is indented to be representative over a broad area and assumes infiltration is added instantaneously, there is no reason to assume that all the soils in California will drain at the same rate. In the revisions, please discuss why this assumption is ok enough for your objectives and how it potentially influences the results.

Indeed, we wholly agree with the remark that the drainage parameter $k_d$ in the AWI model should not be constant across California if one were trying to calibrate such a model to a specific site and specific hydrologic process (such as soil moisture). As properties like soil hydraulic conductivity vary over orders of magnitude, we expect that drainage constants should vary locally. However, as we state in our response to the previous comment, here we are not concerned with calibrating the model to individual sites but instead are normalizing each AWI grid cell by its background climatology (in this case, its 15-year recurrence interval). To try and clarify this up front in our manuscript, we wrote in Lines 184-186, "Whereas constants in eqn. (2) may influence the local magnitude of AWI, for $A^*$ only the relative value is important for a

given grid cell, and for the case of the normalized soil water index, the changing rate constants do not significantly impact the normalized parameter (Osanai et al., 2010)."

For the rate constant $k_d$ used in our model, we somewhat arbitrarily used the value of 0.01/hr from Godt et al. (2006) as it appeared to be able to integrate soil water over multiple days to weeks (their Fig. 4). Although we appeal to previous studies that suggest only the relative rank of peaks in modeled soil water are important for estimating landslide-triggering conditions (Okada, 2005 (in Japanese) as cited in Osanai et al., 2010 (English)), this insightful comment presents a good opportunity to explore the sensitivity of our results to our choice of $k_d$.

With this in mind, the plot below is a modified version of our Fig. 3d where we examine the time series of $A^*$ for our April, 2020 San Diego landslide inventory. Each panel reflects a variation in drainage constant from 0.001/hr (one tenth of our model $k_d$, shown in (a)) to 0.2/hr (20 times greater than our model $k_d$, shown in (d)):

[Figure]

From this plot, we can observe a few different things. First, we see that the two slower drainage cases (a and b) both manage to pick out the landslide event (blue star) well and have respective

$A^*$ values above 1. When $k_d$ values increase to 0.1/hr and above, both the absolute values of $A^*$ decline overall, and the landslide-inducing event does not show a relative increase above over peaks in the time series. This suggests that the increased drainage constant here yields values of $A^*$ that reflect more the high-intensity, short-duration rainfall than the multi-day, long-duration hydrologic evolution from the prolonged cutoff-low storm in April 2020. The decline in absolute value of $A^*$ for the two fast-drainage scenarios (c and d) indicates that the peaks in $A^*$ are not as unique in the climatology when modeled this way.

We see a similar pattern for the case of the 2005 La Conchita time series as well (our original Fig. 3a):

[Figure]

Again, the slow-drainage cases (a) and (b) pick up the 2005 event strongly with $A^*$ peaking above 1. Here, as opposed to San Diego, the faster-drainage case (c) also shows that the 2005 storm is relatively extreme when considering an AWI that more strongly represents high-intensity, short-duration rainfall. For the fastest-drainage case (d), however, the results are similar to the San Diego case where the Jan. 2005 peak in $A^*$ does not capture the landslide event and similar peaks are more prevalent throughout the ~20-year climatology.

From this exercise, it appears that the resultant $A^*$ values do have some sensitivity to the choice of drainage constant, at least for these two documented events. However, ultimately it appears that the choice of $k_d = 0.01$/hr seems to capture the landslide events better than other drainage constants. Perhaps this drainage rate best reflects the balance between long- and short-term rainfall that are relevant to landslide processes, but a more thorough analysis is beyond the scope of this current manuscript. Nevertheless, we thank Reviewer Mirus for this very insightful comment that forced us to more thoroughly consider the choices made in our modeling. We can provide additional context from this new analysis in our revised manuscript as needed.

2. The equation in Godt et al. (2006) accounts for ET in calculating effective rainfall Ii. Did this study also use ET? If not clarify that and explain why. Godt et al. (2006) used a fairly simple monthly estimate of PET, which was good enough for calculating antecedent conditions at the hourly scale, but has a major impact on the outcome if ignored, particularly during spring months when ET ramps up.

Here we do not model E-T as an ongoing process throughout the year, but instead we reset AWI to its residual value at the beginning of each water year to capture the process of summertime evapotranspiration that reduces soil moisture from field capacity down toward its residual moisture content (Lines 182-183). A future version of this model may be improved by incorporating E-T throughout the rainy season, but in our current analysis we do not see any negative impact on our results by not including a continuous E-T process (for example, for our four test inventories we capture two January events, one mid-February event, and one late-April event; Fig. 3).

3. Given that the USGS has published soil moisture data for at least one of the BALT sites with some landslide-inducing storms that you considered (i.e., Thomas et al., USGS-Data, 2017), have you looked at how these compare qualitatively to your AWI calculations for those grid cells? Godt et al. (2006) developed their parameterization of the AWI based on observed soil moisture data from Edmunds, WA (refer to Figure 4 in their paper). With sufficient calibration, one can get an AWI that nicely matches observed soil moisture dynamics for a region of interest (for example, see below).

A primary goal of this research is to determine whether locally extreme values in the hydroclimatology of a universal tank model applied to rainfall data across the state of California act as a reasonable proxy for shallow landslide triggering that can then be used to estimate regions of potential impacts by landslide-inducing storms. Since the output of this tank model can be calibrated to either pore water pressure or soil moisture time series (as discussed in our response on p. 2 of this document), we remain somewhat agnostic as to what process the AWI/tank model specifically captures but instead focus on the fact that it captures some aspect of the current hydrologic state of the system via this mass balance. That being said, we do utilize landslides resulting from the 2017 storms whose data are included in the Thomas et al. (2017) Data Release and show that they correlate with peaks in $A^*$ (e.g., Fig. 2e in the pre-print).

**Data limitations and false/failed alarms**

- The period of record with only 15-years does not adequately support the conclusion of a 15-year recurrence interval for identifying A* and suggests a longer record is needed to better constrain this recurrence-interval based ratio. I didn't quite understand why the full 19 years of the rainfall dataset wasn't leveraged fully.

In this manuscript, we do indeed use the full 19-year record of gridded rainfall (e.g., panel (a) in the figure on p. 5 of this document) to calculate a statewide climatology of modeled soil water using a leaky barrel hydrologic model (panels (b) and (c) in the same figure). Although we would ideally use a longer rainfall record to capture perhaps a more accurate generalized extreme value distribution for each grid cell, one is not available for this dataset which begins in calendar year 2003. We utilize the generalized extreme value distribution approach of Marc et al. (2019) in hopes that we can define the functional form of the hydroclimatology probability distribution given a relatively brief record of gridded rainfall data. However, in the past two decades California has experienced both historic droughts and some of the wettest years on record. This gives us confidence that we are sampling from a broad distribution of climate extremes in our dataset.

- Additionally, although the input rainfall data is time limited, one could easily calculate the AWI going further back with different time-series of rainfall databases to get a sense of whether 15 years is close enough or if much longer records are needed to capture this relevant recurrence interval.

This could certainly be a good exercise for future work, particularly as the NOAA CNRFC gridded QPE dataset we use is limited to the states of California and Nevada (https://www.cnrfc.noaa.gov/arc_search.php?product=netcdfqpe) and other datasets will need to be used to extrapolate this methodology outside of these bounds. Our current results indicate that we are in fact capturing the relevant recurrence interval with this approach. The landslides in our four distinct inventories all plot above a similar recurrence interval of AWI (Fig. 2e in the pre-print), which implies that the methodology yields at least internally consistent results.

- Any criteria used for landslide assessments, whether temporal, spatial, or spatiotemporal should in some way consider the potential for false or failed alarms. I understand that there are data limitations that likely preclude a rigorous quantitative analysis of performance or even explicit consideration of failed alarms or false alarms, but some discussion is needed. Can you identify at least a few "big" storms that might have triggered landslides and did not? Were they also associated with positive (or high) LPA values? Alternatively, from the storms you did evaluate are there any substantial areas with some LPA that *didn't* produce any landslides (probably, given your very conservative 5-degree slope threshold), and if so, how much?

For the regional approach we are taking, the four landslide inventories used in this analysis are unfortunately insufficiently broad to capture spatial gradients in $A^*$ for a given event that can be used to evaluate the efficacy of $A^*$ in producing false positives and false negatives. Because we

could not do this in the spatial dimension for a given storm with our current inventories, we instead opted to show this qualitatively in the time dimension by looking at the 19-year time series of $A^*$ for a bounding box at each landslide inventory (Fig. 3 in the manuscript). We note in lines 358-367 that because detailed histories are not available at each box it is difficult to evaluate false positives and negatives, although there are certain cases (like the East Bay hills) where we can be fairly confident that a significant regional landslide event like what was observed in 2017 had not occurred since the winter of 1997-1998 (e.g., Coe and Godt, 2001).

The other approach we take toward discussing the uncertainty in $A^*$ as a true classifier of landslide and non-landslide events is an evaluation of recent major atmospheric river storms that impacted California in January 2023 (Fig. 8 in the pre-print). Here we use rainfall data that postdates the dataset used to calculate $A^*$ to assess how our calibrated threshold works for a recent series of storms where landslides were reported across the state. In our discussion of these results, we write (in lines 535-546):

> Figure 8 shows snapshots of $A_*$ maxima and reported landslides from the CGS database during two of the most intense storm periods during the 2023 AR sequence: 30 December 2022 – 03 January 2023 (AR3; Fig. 8a), and 09 January – 11 January 2023 (AR3; Fig. 8b). Overall, the footprints of $A_*$ generally cover the zones of high landslide density at the regional scale for both cases. Isolated landslides are not very well-resolved by the method; however, some events in the reported landslide database may be related to roadcut failures and may not reflect purely natural conditions. Additionally, at this level of mapping it is difficult to evaluate false positives (i.e., zones of above-threshold $A_*$ where reported landslides are absent) because of potential reporting biases. For example, landslides may be under-reported in areas of low road or population density, or in instances when certain roads may have already been closed due to storm damage. Future work with a more robust mapping of natural failures across the entire domain (a rare and comprehensive, time-consuming effort) would help quantify prediction uncertainty. Even so, a gross comparison of reported landslide spatial density with increases in $A_*$ (Fig. 8c) shows a marked rise in landslide kernel density as $A_*$ approaches and exceeds a value of 1, our calibrated threshold based on the local 15-year recurrence of $AWI$.

In the above paragraph, we mention how from this analysis our $A^*$ metric is not a good predictor for single-event or very low-spatial-density landsliding (e.g., Fig. 8c), but appears to correlate well with areas that experienced more regionally dense slope failures. We can add more discussion here, as one can see from Fig. 8a that there are areas of over-threshold $A^*$ from the Dec. 30 – Jan. 3 storm along the Sierra Nevada (purple area that intersects the 120ºW longitudinal meridian) where landslides were not reported. Again, it is unclear how much this is due to inaccuracy in the model or incomplete reporting, but it is certainly worth discussing. Future work will incorporate a more thorough evaluation of the quantitative efficacy of $A^*$ as a predictor of landslides.

Lastly, although we mostly focus on landslide-inducing storms and a more thorough analysis of null events remains out of the scope of this study, we do present a historically large AR5 storm that made landfall in October 21 and did not result in regionally distributed landsliding. In Fig. 6 we show that although the two-day rainfall totals were anomalously high, the resulting A* was low due to the dry soils at the storm onset.

- Can you correlate the LPA to landslide densities, numbers of landslides, or even just a general qualitative descriptor of how widespread landslides were within the areas predicted? Does a higher LPA link to a greater number/severity of landsliding? I ask this because the magnitude varies so widely and the slope is so low, that it seems some guidance would be needed to understand what the difference might be between LPA = 60 vs. LPA = 11k. Given the conservative slope threshold, I could easily imagine a scenario in which lower LPA in a localized area of very steep terrain triggers a higher density of landslides than a broader area of less steep terrain with a much higher total LPA.

We very much appreciate this comment and it is one we have thought about extensively! Landslide spatial density does indeed appear to increase nonlinearly as $A^*$ approaches and exceeds a value of 1, as we show in Fig. 8c (and show a revised version using actual landslide counts in our response to Reviewer Marc's comments). This is not reflected in the value of LPA, which is purely a measure of hillslope area experiencing above-threshold conditions. We feel that LPA is a useful construct in and of itself; however, in a previous iteration of this work we took the idea one step further and created an integrated quantity that reflected both the impacted area as well as the intensity of $A^*$ (which we called the landslide storm magnitude, or LSM) but this was not looked upon favorably by prior reviewers perhaps as we had not yet conducted the analysis of the 2023 storms and their landslide distributions that show a relationship between landslide density and $A^*$ (Fig. 8). In our response to Reviewer Marc's comments regarding Fig. 8c, we now have a coarse quantitative relationship between relative landslide density and $A^*$ (see our other response document for an updated Figure). If a researcher or emergency manager were to want guidance on the local landslide intensity potential for a given storm, this power-law relationship could be used with a map of $A^*$. In the future we hope to better incorporate the impact of variable $A^*$ on an integrated metric of landslide storm magnitude.

Regarding slope, we chose a 5-degree cutoff for the 30-m DEM used in the analysis of LPA based on the distributed of observed slopes for our landslide inventory (see histogram below). Here we sampled the underlying 30-m slope pixel of each landslide point of our four inventories, which is a likely underestimate of the true slope as the landslides themselves are often much smaller in aerial extent:

[Figure]

Although a five-degree cutoff may be somewhat conservative, this is only an estimate of the apparent (downsampled) slope and it is within two standard deviations of the observed mean. It is also interesting to note that the slope distribution itself is quite normally distributed (albeit with a slight skew) and therefore "very steep terrain" may not correlate well with the most likely area for the types of shallow landslides modeled in this work. This is similar to what is observed in studies like Prancevic et al. (2020). Therefore, future modeling of landslide triggering by $A^*$ may benefit from incorporating this Gaussian relationship between underlying slope and probability. However, we found this out of the scope of the current study.

**Specific line-by-line comments:**

12 – Can you provide a few words that better characterize A* for the abstract? Is it a proxy, or a dimensionless index that you test as a proxy for landslide probability? It's not clear until I read the paper.

Certainly, we can adjust the language in the abstract in our potential revision to include your suggestions. We will re-word this sentence to say, "As mapped landslide inventories typically

cover a small region relative to a storm system, here we develop a dimensionless index based on locally anomalous soil water, $A^*$, that we test as a proxy for shallow landslide triggering."

54 – Consider mentioning however, that most of those ARs from their analysis did NOT result in landslide triggering. So, there's more going on than just high IVT and ARs.

Thank you for bringing this up at is a crucial point that helps motivate our work in this manuscript. We will be sure to include this in our revision.

87 – I'm not sure that this Bogaard and Greco paper is an appropriate citation for the ID threshold since it largely focuses on the inconstancies and advocates for moving away from the approach.

Yes, this reference indeed is inappropriate here and will be removed in our revision.

115 – It's worth citing some of the other recent literature using direct measurements of soil water content to identify empirical thresholds (see Mirus et al., NHESS Discussion, 2024 and references therein).

We can certainly include some more recent references on hydrometeorological thresholds for landslide triggering as described in Mirus et al. (2024), particularly the discussion on the bilinear threshold methodology that represents a more empirical approach to threshold determination than the deterministic methodology of Thomas et al. (2018) that we reference within this paragraph of the manuscript.

156 – Technically this isn't a parameter, but rather an index or nondimensional ratio that is the output of your calibrated/parameterized AWI model.

Fair enough, we will be sure to re-word language throughout the text to ensure that the proper terminology is used to describe $A^*$.

178 – Godt et al. (2006) used an effective rainfall minus PET to calculate Ii, so that's an important consideration especially for California, particularly to consider potential impacts of climate change.

As we describe above in our answer at the top of p. 9, in our model we compensate for E-T annually by resetting the model at the beginning of each water year to its residual condition.

195 – Consider the Thomas et al., (*WRR,* 2019) not only showed that SMAP over-smoothed the soil moisture data, but PRISM also over-smoothed out the orographic effect on precipitation intensity and totals. Your dimensionless approach with A* should help avoid this issue of magnitude of precipitation (or soil moisture) measurements, which is a common issue with satellite precipitation estimates.

Yes, in our initial evaluation of gridded quantitative precipitation estimates we found that a gauge-interpolated dataset for our region seemed to work quite well, although we have yet to do a thorough sensitivity analysis with different satellite- or terrestrial-based rainfall datasets.

203 – Again, 15-years seems like a short window considering your conclusion and that Marc et al. (2022) found that the 10-year rainfall anomaly was potentially only 50% effective in characterizing landslide potential for extreme and widespread landslide events. Some discussion or further analysis using a proxy with older rainfall records for one (or more) location seems prudent to better support your conclusion.

In our response to a similar summary comment, we note that unfortunately a coding error resulted in an artificial increase in AWI and thus the threshold recurrence value. Our corrected threshold recurrence interval is actually 10 years (see updated plot in response to Reviewer Marc comments), which is likely better captured in our rainfall climatology. As we also state above, in between Water Years 2004-2022 California experienced historic climatological extremes, and therefore our associated AWI extreme value distributions should reflect this broad representation of California's hydroclimatology.

It is important to note that our method diverges from that of Marc et al. (2019, 2022) in the use of a continuous soil water mass balance rather than a 2-day rainfall anomaly. Although these excellent studies inspired us to explore an approach that utilized a spatial normalization process to account for climate-dependent impacts on local geomorphology which obviates the need to calibrate site-specific parameters, we knew that at least in California we needed some way to consider soil storage and memory effects that impact landslide triggering particularly for early-season storms. That is what led us to explore hydrologic models that are simple to employ yet capture the essential dynamics for shallow landslide triggering. In our manuscript, we included a comparison between our hydrologic model approach ($A^*$) vs. the two-day rainfall anomaly ($R^*_{48}$) approach of Marc et al. (2019, 2022) on a historic extreme early-season storm that did not produce widespread landslides in California (Fig. 6). Here the two-day rainfall anomaly ($R^*_{48}$) is quite high across a band of Northern California (Fig. 6b), yet the $A^*$ values are well below 1 (our identified threshold) due to the initially dry conditions before the arrival of the storm (Fig. 6c).

Additionally, in Marc et al. (2022) the authors seek to test satellite multisensor precipitation products (SMPPs) for use with their $R^*$ methodology as opposed to the interpolated gauge data utilized in their initial 2019 manuscript. Here the authors find that the SMPPs in many cases do not correctly capture associated landslide locations. In the last paragraph of Section 4.c.1 the authors write, "…in many of the studied cases, the poor link between SMPP retrieval and landslide location is likely due to inadequate rainfall retrieval from the SMPP rather than due to some independent complexities in the landscape and/or the landslide mapping. In the next section we briefly identify options to improve the SMPP algorithms." Thus, for a number of reasons it is not a clear apples-to-apples comparison between our methodology and results to those of Marc et al. (2022).

207 – Technically you're not calibrating A* since there isn't any parameter adjustment, you're identifying which value of A* corresponds to landslide occurrence in the same way one might use a critical factor of safety for estimating slope failures.

This is a fair point and we will be sure to correct our language here.

Also, refer to general comment about kd. Was the soil variability considered in the field capacity, etc. considered for calibrating kd in area without monitoring data or landslide occurrence data?

See our response to this summary comment on page 6.

223 – Consider specifying how the interpolation is applied (linear? Krigging? Other?) rather than citing a toolbox and assuming everyone knows how it works.

Apologies, we will be sure to describe more clearly the linear interpolation regime used in this tool.

225 – Why not use a 19-year time series to identify the anomaly?

We do indeed use the 19-year time series of $A^*$ here to show how the peak value across the 19-year record corresponds to the general timing of the landslide-inducing storm (Fig. 3). In Lines 224-226 we write, "…we also examined the 19-year time series of $A^*$ across each inventory to illustrate the unique occurrence of these values throughout each of the four calibration storms."

236 – Why not use a more sophisticated approach than 5-degree slope? For example, the Godt et al., *NASL*, 2012) model is available at 1km resolution and Brabb et al, *USGS,* 1999 found a 25degree slope was suitable for rapid shallow landslides and debris flows. Within this context your threshold is very conservative. Discuss?

In this work we are focused on delineating the broad areas that are experiencing hydrologic conditions associated with shallow-landslide-triggering and are not attempting to perform a more comprehensive stability assessment that explicitly considers other physical factors such as topographic slope, soil strength, or root reinforcement. Instead, we opt to separate hillslopes from flat areas based on the distribution of observed slopes from our four landslide inventories (see Figure on p. 12 of this response). In our case, from the 30m DEM used to calculate LPA we find that an apparent 5-degree cutoff captures most of the landslides in our datasets (see histogram above of landslide apparent slopes). In our future work we plan to incorporate $A^*$ into a more comprehensive predictive model that may incorporate the Gaussian slope distribution portrayed by our landslide data.

Figure 2. This relationship does not appear to be 1:1 and suggests that your AWI might need to vary for different landslide triggering storms or locations (see general comment #2 about calibrating kd). How do your other 5 storms plot here?

This is a good observation, but given we only have four landslide inventories it is not totally clear whether this trend would hold for other events. Although we would love to plot our other

events on here, we only have good inventories for the four events in Fig. 2. Thus, we do not attempt to calibrate a more complex threshold.

423 – Do you mean Luna and Korup, 2022?

Correct! Thank you for this catch.

247 – What about false alarms?

In this analysis we considered the October 2021 storm a relative false alarm as it had anomalously high 2-day rainfall totals but no instances of widespread landsliding (as we write in line 251). However, we could certainly be more clear on this front and will correct the language to reflect our deliberate choice of testing our metric on what may be considered a false alarm depending on the metric for landslide triggering utilized.

266-67 – I don't quite follow this. Isn't the maximum AR value used going to be independent of location to produce the value and time of AR max? How does it correspond to the location of one or more observed landslides?

We should clarify that in nearly all cases the highest AR-scale value is indeed regionally close to the zone of known landslides, and we can include supplementary data showing more about the AR scale estimation. Calculating AR scale farther inland is less indicative of landfalling storm strength as much of the IVT signal, which represents the vertical integral of winds and water vapor, declines inland from the coast. Therefore, relating AR scale at the exact landslide location may not reflect the true nature of the landfalling storm, and that is why we calculated AR scale using the standard meteorological approach (as described in lines 263-267).

Table 1 – This includes results and should probably be linked to those later in the paper. Also, can you list the number or density of landslides you showed in Figure 8c (albeit, maybe acknowledging that the inventories are incomplete)?

We will be sure to reference Table 1 later in the manuscript. Regarding Fig. 8c, it is difficult to translate this value to a true landslide spatial density with this approach as we used a Gaussian kernel. In a revised figure (see response to Reviewer Marc comments), we instead take a grid approach to calculate the true landslide numbers within 15-arc-minute grid cells.

What about non-events? Are you able to introduce any of these, even anecdotally?

The October 2021 storm described above is one rather sizeable non-event in our catalog (e.g., Fig. 6), although isolated instances of landslides did occur. A future study could more explicitly compare our catalog to more non-events, although this is difficult to achieve at a statewide level given the general lack of landslide datasets available which makes it uncertain whether or not landslides occurred at a given place and time.

Figure 4 – This is a European/international journal, so I suggest you clarify that the numbers are years (e.g., January 2005, not January 5[th])

Thank you for pointing this out! We will include the full year in the figure panels to avoid any misconception regarding event date.

453 – Again, a good place to point out that Cordeira found that while most landslides were associated with ARs, most ARs didn't necessarily produce landslides.

Agreed, and we will be sure to mention this observation in our revision.

465 – Maybe I missed it, but your dataset considered 6h rainfall, so how does that account for the high-intensity bursts?

In the cases of Fig. 7a and 7b, the rainfall bursts apparently were sufficiently large or long-lived to show up in the 6-hourly data and are reflected in the resultant $A^*$ values. That being said, it would not be expected for sub-hourly intensities to necessarily be reflected in the 6h data. As we note in the discussion, however, for the case of the March 2018 storm the passage of NCFR's was not precisely captured in the gauge-interpolated gridded rainfall dataset yet show up nicely on radar (Fig. 6b; Lines 495-500). In response to Reviewer Marc's comments, we will also include maps of resultant $A^*$ values within this figure.

Fig 8c. This is the most compelling argument for the broad utility of this approach. You might be able to do even better if your kd values varied by soil type and ET were considered in calculation of AWI.

Perhaps! Although we present arguments within this response document for the utility of normalizing a universal model and deciding how the drainage coefficient varies with soil type presents its own challenge, these are out of the scope of the work presented but we do agree that future work could include a more thorough sensitivity analysis of how spatially variable parameters may help even a hydroclimatologically normalized index.

540 – Here or elsewhere warrants some discussion of false positives and false negatives, which are not tested thoroughly. Even if this is justified by a lack of comprehensive data, this is still a limitation of the approach worth discussing (if only to point out the data needs for further developing this method!).

We can certainly flesh out our discussion here in lines 535-546 that describes the apparent shortfalls of the model applied to the 2023 storms and difficulties in evaluating false positives and false negatives.

548 – Again, what about local variation in soil properties (depth, ksat, retention curve, etc.) all of which are wrapped up in the kd parameter that you seem to have held constant for the entire State of California?!

We discuss the advantages of normalizing AWI by a representation of its local climatology (i.e., looking at $A^*$ instead of AWI), and the impact of variable $k_d$ on $A^*$ predictions in pages 6-8 of this response document.

569 – This potential value underscores the need to consider ET in the effective rainfall Ii.

We discuss our choices in approximating E-T on p. 9 of this response document, and we agree that future studies may benefit from a more statewide explicit modeling of E-T.